# Metal-dependent and metal-free mechanisms of peptide condensate catalysts

Tlalit Massarano [1,10], Yuqin Yang [2,3,4,5,6,10], Avigail Baruch Leshem[1], Ori Eran[1], Xiaoyu Wang[2,3,4,5,6], Hao Dong [2,3,4,5,6] ✉ & Ayala Lampel [1,7,8,9] ✉

Condensates formed via liquid-liquid phase separation (LLPS) provide a chemically versatile environment for catalysis through dynamic molecular interactions. We present designed biomolecular condensates, formed by LLPS of minimalistic histidine-containing peptides, catalyzing ester hydrolysis with two distinct mechanisms. $Zn^{2+}$-dependent condensates activate a coordinating water molecule at the active site, formed by $Zn^{2+}$-histidine coordination, enabling nucleophilic attack. We show that dense-phase basicity, internal mobility, and $Zn^{2+}$ accumulation within the condensates collectively govern their catalytic activity. In the absence of $Zn^{2+}$, catalysis is driven by intermolecular low-barrier hydrogen bonds between histidine residues, facilitating nucleophile formation. Combined computational and experimental evidence reveals the molecular basis of these catalytic pathways, demonstrating the functionality of biomolecular condensates in catalysis and nanotechnology. These findings establish a foundation for exploring mechanisms of metal-free emergent catalysis within complex liquid assemblies, expanding the potential of LLPS-based systems in green chemistry and advanced materials.

Enzymes facilitate a diverse range of reactions under mild aqueous conditions with exceptional efficiency and specificity. Drawing inspiration from enzymes, researchers have developed supramolecular catalysts[1], utilizing self-assembled building blocks[2,3] to construct complex structures with emergent and tunable properties[4–6]. Peptides have gained prominence as supramolecular catalytic building blocks due to their biocompatibility, natural presence in enzymatic catalytic sites, and chemical versatility[2,4,5,7]. Peptide-based self-assembled functional materials, including α-helix[8], hydrogels[9]/fibrils[10,11], nanotubes[12] and nanowire superstructure[13] have been developed as artificial biocatalysts[2–5,7,10,12,14–17]. These assemblies mimic diverse enzymatic capabilities, including peroxidase[18], aldolase[19]/retro aldolase[20], and

hydrolase[11,16] activities. Hydrolases, particularly esterase- and phosphatase-mimetic supramolecular systems, have been extensively studied for their applications in drug delivery, agriculture, and environmental remediation[15].

The active sites of hydrolases frequently contain histidine (His) residues, which play a central role in catalysis. His may function as ligands for metal ion cofactors, participate in catalytic dyads or triads, or mediate reactions through interactions with other His residues[21]. In esterase mimics, His residues often coordinate $Zn^{2+}$ ions, providing structural stability and catalyzing ester hydrolysis[13,16,22]. Computational studies have shown that three His residues can coordinate $Zn^{2+}$ ions, with a fourth His residue accepting the excess proton[23,24]. Alternatively,

[1]Shmunis School of Biomedicine and Cancer Research, George S. Wise Faculty of Life Sciences, Tel Aviv University, Tel Aviv, Israel. [2]State Key Laboratory of Analytical Chemistry for Life Science, Nanjing University, Nanjing, China. [3]Kuang Yaming Honors School, Nanjing University, Nanjing, China. [4]Chemistry and Biomedicine Innovation Center (ChemBIC), Nanjing University, Nanjing, China. [5]ChemBioMed Interdisciplinary Research Center, Nanjing University, Nanjing, China. [6]Institute for Brain Sciences, Nanjing University, Nanjing, China. [7]Center for Nanoscience and Nanotechnology, Tel Aviv University, Tel Aviv, Israel. [8]Center for the Physics and Chemistry of Living Systems, Tel Aviv University, Tel Aviv, Israel. [9]Leibniz Institute of Polymer Research Dresden Max Bergmann Center of Biomaterials Dresden, Dresden, Germany. [10]These authors contributed equally: Tlalit Massarano, Yuqin Yang. ✉e-mail: donghao@nju.edu.cn; ayalalampel@tauex.tau.ac.il

His residues can catalyze ester hydrolysis without $Zn^{2+}$ ions[10,11,25], wherein adjacent His residues are both protonated and deprotonated participate in the catalysis[26].

Moving from hierarchical supramolecular structures to disordered liquid assemblies, the emergent field of liquid-liquid phase separation[27] (LLPS) provides opportunities to design dynamic and tunable environments. Inspired by membraneless organelles[28], biomolecular condensates are dense phases formed via LLPS of biomolecules[27], driven by multivalent, weak non-covalent interactions. A decade of research into the amino acid or nucleic acid interaction modules underlying LLPS[27,29] has enabled the design of condensates[30] based on various building blocks, including engineered proteins and polypeptides[31], DNA[32]/RNA[33] and short peptides[34-37]. The chemical diversity and sequence programmability of designed condensates allow precise control over their formation and properties.

In addition to their ability to sequester a broad range of molecules, including large macromolecules and small hydrophobic compounds, condensates have been employed in various applications mimicking cellular organelles, such as microreactors[37-41], cell-free transcription-translation systems[42], and intracellular delivery[43]. More recently, the spatiotemporal regulation of chemical reactions by condensates suggest their potential to function not only as vessels for chemical reactions[37,39,44] but also as active catalytic entities[45-49].

Designing condensates as catalysts can be achieved through various strategies, including the fusion of enzymes to low complexity domains (LCDs) that promote phase separation[49], utilizing intrinsically disordered proteins[47,48], employing short peptides alone[46] or with other building blocks[45]. These condensates enhance catalysis through interfacial electric potentials[47,48], distinct intrinsic environment[45,46,49] and partitioning[46]. Despite these advancements, many questions remained unanswered about the molecular mechanisms underlying the catalytic capabilities of condensates.

Here, we introduce a class of catalytic condensates formed through complex coacervation of His-containing peptides. The formation of condensates, along with their catalytic ability to enhance ester hydrolysis, are mediated by two independent pathways: a metal-dependent mechanism and a metal-free mechanism. In the metal-dependent pathway, $Zn^{2+}$ ions coordinate with the peptides and activate their bound water molecules to generate nucleophilic hydroxide groups, while in the metal-free pathway, intermolecular low-barrier hydrogen bonds (LBHB) between His residues enhance nucleophile formation.

Our findings shed light on the molecular mechanisms underlying catalytic condensates and connect the condensates materials properties with their catalytic capacity emphasize the potential of phase-separating liquid materials in regulating reactivity. Furthermore, they highlight promising opportunities for designing supramolecular liquid-based biocatalysts through precise molecular engineering, a process expected to be further accelerated with the integration of machine learning[50] tools in the near future.

## Results

### Peptide LLPS is mediated through coordination to $Zn^{2+}$ ions

We sought to develop a minimalistic condensate system with catalytic capacity by utilizing LLPS-promoting peptides through the incorporation of His as catalytic moieties. We hypothesized that these condensates could enhance ester hydrolysis due to their dynamic character and His-rich catalytic site. We proposed that the dense phase of the condensates might facilitate substrate partitioning and provides a molecularly enriched environment that enables multiple peptide-peptide interactions, in which catalytic moieties promote the substrate hydrolysis (Fig. 1a). The condensates are formed by two oppositely charged peptides, which builds on our recent design of minimalistic LLPS-promoting peptides that assemble into biomolecular condensates[36]. We incorporated either tyrosine (Tyr) or one or

two His, separated by Gly, at the C-terminus of the peptide to generate a small library of His-containing LLPS-promoting peptides (Fig. 1b).

We analyzed the LLPS propensity of oppositely charged peptides with varying number of His: R2H/E2H, R1H/E1H and R/E (Fig. 1). Building on previous work on esterase-mimicking catalytic peptides[16], we used $Zn^{2+}$ as a cofactor for ester hydrolysis and hypothesized that $Zn^{2+}$ could function both as a cofactor and an initiator of LLPS (Fig. 1). We created phase diagram heat maps by monitoring sample turbidity as a factor of peptide concentration (Fig. 2a–c, Supplementary Tables 2–4) and confirmed condensate formation by bright-field microscopy (Supplementary Fig. 1).

Phase diagrams showed an inverse relationship between the number of His residues and the saturation concentration ($C_{sat}$, Supplementary Fig. 1), suggesting that His enhances LLPS through specific interactions with $Zn^{2+}$ ions. Replacement of $ZnCl_2$ with $CaCl_2$ or $NaCl$ confirmed that LLPS is driven by $Zn^{2+}$ coordination, alongside peptide-peptide interactions (Supplementary Fig. 2). Previous studies also support the role of divalent metal ions, particularly $Zn^{2+}$, in mediating phase separation[51,52]. Moreover, the effect of $ZnCl_2$ on LLPS is concentration-dependent (Supplementary Fig. 2), where 0.2 mM is the critical concentration to trigger LLPS. Thus, $ZnCl_2$ concentration directly modulates the dense-phase concentration of the R2H/E2H peptides. Increasing the $ZnCl_2$ concentration from 0.2 and 0.33 mM to 0.67 mM resulted in 1.7-fold and 2.2-fold increases, respectively, in the total peptide concentration within the dense phase (Fig. 2d). Dense-phase peptide concentrations were determined by quantifying the peptide concentration remaining in the dilute phase following droplet centrifugation, using absorbance spectroscopy calibrated for the peptides. These findings indicate that $Zn^{2+}$ concentration may also play a critical role in influencing the catalytic capacity of the condensates.

### The catalytic performance of the condensates is restricted by $Zn^{2+}$ ions

We aimed to analyze the catalytic capacity of the peptide condensates, focusing on the His-rich R2H/E2H system at 0.5 mM. To this end, we selected 4-methylumbelliferyl acetate (4-MU-Ac) as the model ester substrate for hydrolysis (Supplementary Fig. 3), which produces the fluorescent product 4-methylumbelliferone (4-MU). Due to the low solubility of 4-MU-Ac and 4-MU in aqueous solutions, the selected concentration range was limited to 1.1 mM, resulting in linear initial velocity ($V_0$) plots (Fig. 2e). To consider the condensates' impact on 4-MU's fluorescence intensity, we used calibration curves in condensate solution (Supplementary Fig. 4) and confirmed that 2.8% MeCN, used to dissolve 4-MU-Ac, did not affect condensate formation (Supplementary Fig. 5).

We evaluated the catalytic performance and product formation (Supplementary Fig. 6) in the presence of $ZnCl_2$ (0.67 mM). The $V_0$ value ($2.0E-05 \pm 7.0E-5$) was found >6.5-fold higher than that in bulk buffer in the presence of $Zn^{2+}$ at substrate concentration of 0.7 mM (Supplementary Table 5). Each of the peptides independently, R2H and E2H, does not undergo LLPS yet affect the reaction kinetics, where the $V_0$ is ~2-fold and 3.7-fold faster under LLPS conditions (R2H/E2H) compared to when no LLPS is observed (R2H or E2H alone) (Fig. 2e, Supplementary Table 5). Notably, dynamic light scattering (DLS) analysis showed that the soluble R2H and E2H peptides form nanometer-sized assemblies (Supplementary Fig. 7). The condensates formed following complexation of R2H and E2H have an average size of $1.2 \pm 0.25$ µm. This observed $V_0$ value of R2H/E2H with 0.67 mM $Zn^{2+}$ is lower than those previously reported for $Zn^{2+}$-dependent self-assembling catalysts with ordered structures for p-nitrophenyl acetate (pNPA) hydrolysis[3,16,22]. Yet, decreasing the $Zn^{2+}$ concentration to 0.33 mM and 0.20 mM resulted in 2.6-fold and 3.4-fold increase in the $V_0$ at 0.7 mM 4-MU-Ac compared to 0.67 mM $Zn^{2+}$, respectively (Fig. 2e). This unexpected trend was also observed in a previous study[53], however, most studies on metal-dependent self-assembled

peptide catalysts report on acceleration of reaction kinetics mediated by Zn²⁺ ions[16,22].

To further elucidate how Zn²⁺ concentration influences the catalytic activity of the phase-separated systems, we measured the pH of both the dilute and dense phases using the ratiometric pH probe SNARF-1 and by confocal laser scanning microscopy (CLSM)[48,54,55]. The emission ratio of SNARF-1 at $\lambda_{em}$ = 580 and 640 nm varies as a function of pH (Supplementary Fig. 8). Calibration curves were obtained by measuring the $I_{580/640}$ ratio in buffers of defined pH containing varying Zn²⁺ concentrations, enabling quantitative determination of pH in each phase (Fig. 2f–g). The pH of the dilute phase was similar across all systems, with 7.7 ± 0.11 and 7.6 ± 0.40 for 0.2 mM and 0.67 mM Zn²⁺, respectively. In contrast, the dense-phase pH was strongly dependent on Zn²⁺ concentration, decreasing from a highly basic value of 10.8 ± 0.60 at 0.2 mM Zn²⁺ to 8.5 ± 0.10 at 0.67 mM Zn²⁺ (Fig. 2f–g, and Supplementary Table 6). We further analyzed how Zn²⁺ concentration affect the internal mobility of the dense phase using fluorescence recovery after photobleaching (FRAP) by CLSM. For this, we used rhodamine b as a fluorescent payload and probe. By tracking the recovery of rhodamine b fluorescence following photobleaching

(Fig. 2h–j) we obtained the $t_{1/2}$ values for condensates with increasing Zn²⁺ concentrations (Fig. 2j). While all three systems exhibit condensate coalescence characteristic of LLPS (Supplementary Videos 1-3), increasing Zn²⁺ concentration leads to distinct differences in diffusivity. We note that the absolute diffusivity values obtained using rhodamine b may be influenced by electrostatic dye–peptide interactions. At 0.67 mM Zn²⁺, only 35.5 ± 11.7% recovery of rhodamine b signal was observed, indicating arrested solid-like dense phase with restricted diffusion (Fig. 2h-i). Lowering the Zn²⁺ concentration to 0.33 mM and 0.2 mM, resulted in increase in the recovery after photobleaching to 81.1 ± 6.3 % and 84.9 ± 5.2%, respectively (Fig. 2i), with $t_{1/2}$ values of 48.8 ± 7.7 s and 36.0 ± 5.0 s for 0.33 mM and 0.2 mM Zn²⁺ (Fig. 2j). Additionally, we analyzed the Zn content within the dense phase of droplets formed at 0.2, 0.33, and 0.67 mM ZnCl₂ using scanning transmission electron microscopy (STEM) coupled with energy-dispersive X-ray spectroscopy (EDS). High-angle annular dark-field (HAADF) imaging combined with EDS line profiling enabled comparison of the relative Zn signal within the dense phase (Fig. 2k). This analysis revealed a pronounced enrichment of Zn at higher Zn²⁺ total concentrations, with ~1.5-fold and 5-fold increases in Zn intensity in

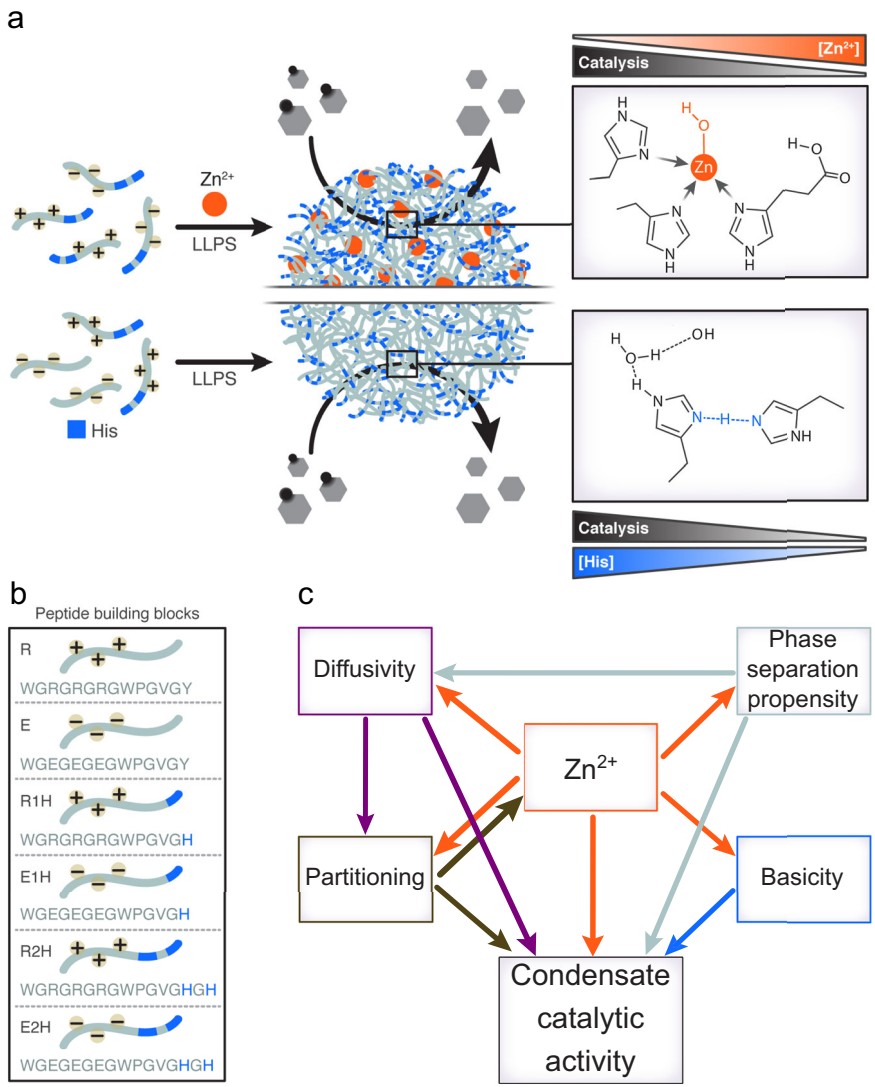

**Fig. 1 | Designer biomolecular condensate catalysts. a** Schematic illustration of condensate catalysts that enhance ester hydrolysis. The formation of the condensates and their catalytic capacity is mediated by two independent pathways, through Zn²⁺-dependent mechanism or Zn²⁺-free mechanism. In the first mechanism, Zn²⁺ ions mediate LLPS and nucleophile formation, where they restrict the catalytic activity of the condensates. In the second mechanism, which is mediated by intermolecular interactions between His residues, increasing number of His residues enhance catalysis. **b** Peptide building block library. **c** Summarizing schematic framework showing the relations between the properties of the designed condensates and their effect on catalysis.

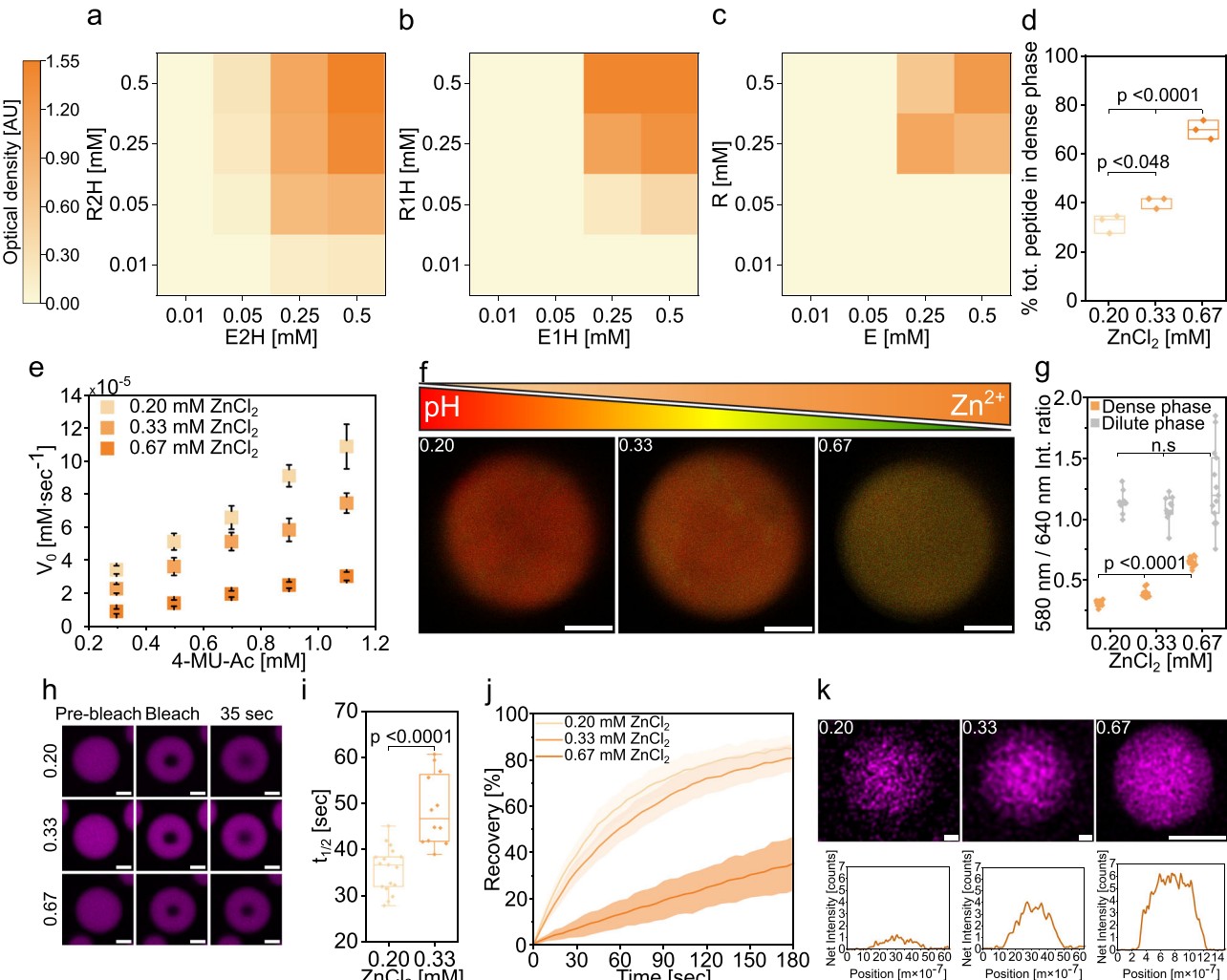

**Fig. 2 | Zn²⁺-dependent catalytic activity and physical properties of peptide condensates.** **a**–**c** Phase diagram heat maps showing optical density (arbitrary units, AU) of Zn²⁺-dependent biomolecular condensates formed by (**a**) R2H/E2H, (**b**) R1H/E1H, and (**c**) R/E in 10 mM Tris-HCl buffer with 0.67 mM ZnCl₂, pH=7.5. **d** % total peptide in the dense phase of R2H/E2H condensates at varying Zn²⁺ (Tris buffer, pH 7.5). Individual data points are shown. Two-sided *p* values: *p* = 0.04802 for 0.33 mM vs 0.20 mM; *p* < 0.0001 for 0.67 mM vs 0.20 mM and 0.67 mM vs 0.33 mM. **e** Kinetics of 4-MU-Ac hydrolysis catalyzed by 0.5 mM R2H/E2H with 0.67, 0.33 and 0.20 mM ZnCl₂ (Tris buffer, pH 7.5). Values represent averages, error bars represent S.D. **f**, **g** CLSM analysis of the pH in the dense phase using SNARF-1 (20 µM) for R2H/E2H condensates at varying Zn²⁺ concentration (Tris buffer, pH 7.5) **f** Representative CLSM images, scale bars=2 µm. The experiment was performed once under the conditions described. *n* numbers are indicated in Supplementary Table 6. **g** Box plots of the SNARF-1 580/640 nm emission intensity ratio measured in the dense and dilute phases of R2H/E2H condensates at increasing Zn²⁺ concentrations. The experiment was performed once under the conditions described. *n* numbers are indicated in Supplementary Table 6.

Individual data points are shown, each represent a single condensate or dilute area. Two-sided *p* values: dilute- *p* = 0.38579 for 0.33 mM vs 0.20 mM; *p* = 0.37218 for 0.67 mM vs 0.20 mM; *p* = 0.15682 for 0.67 mM vs 0.33 mM. All *p* values are *p* < 0.0001 for condensates. **h-j**. FRAP CLSM analysis of R2H/E2H condensates at varying Zn²⁺ concentration (Tris buffer, pH 7.5), performed using rhodamine b (0.1 µM) as a fluorescent probe (λ_ex = 560 nm). **h** CLSM images of condensates before, immediately after and 35 sec after bleaching. Scale bars=2 µm. The experiment was independently repeated three times for the 0.20 and 0.67 mM ZnCl₂ conditions and twice for the 0.33 mM ZnCl₂ condition. **i** Calculated t_{1/2} values. Individual data points are shown. Two-sided *p* values: *p* < 0.0001. **j** Recovery plots, shaded areas represent ± S.D. **k** STEM-EDS elemental mapping and corresponding net intensity profiles showing Zn distribution within R2H/E2H condensates. Scale bars=500 nm. Box hinges indicate the first and third quartiles of the corresponding data, mid lines represent medians, and whiskers span the range of the data. See Supplementary Table 11 for details on statistical analyses. Source data are provided as a Source Data file.

droplets formed at 0.67 mM Zn²⁺ compared to those formed at 0.33 mM and 0.2 mM Zn²⁺, respectively (Fig. 2k). The partitioning of the 4-MU product was weakest in condensates formed at 0.67 mM Zn²⁺ (Supplementary Fig. 9), likely due to the highly crosslinked dense-phase network at this Zn²⁺ concentration, which reduces accessibility and limits internal porosity.

The difference in dense-phase pH at varying Zn²⁺ concentration can be attributed to Zn²⁺-His coordination. At lower Zn²⁺ concentrations, fewer His residues are coordinated, increasing the effective basicity of the dense phase. At higher Zn²⁺ concentrations, extensive

Zn²⁺-His coordination reduces the availability and basicity of free His residues, resulting in a lower dense-phase pH. The elevated basicity observed at low Zn²⁺ is expected to promote the hydrolytic reaction, consistent with enhanced reaction rate measured in bulk buffer under basic conditions (Supplementary Fig. 10) and provides a mechanistic basis for the highest catalytic activity observed at 0.2 mM Zn²⁺. FRAP measurements further demonstrate that increasing Zn²⁺ markedly attenuate molecular diffusivity within the dense phase, likely due to Zn²⁺-mediated physical crosslinking of the peptide network. This limited diffusion and mass transfer at higher Zn²⁺ concentrations can

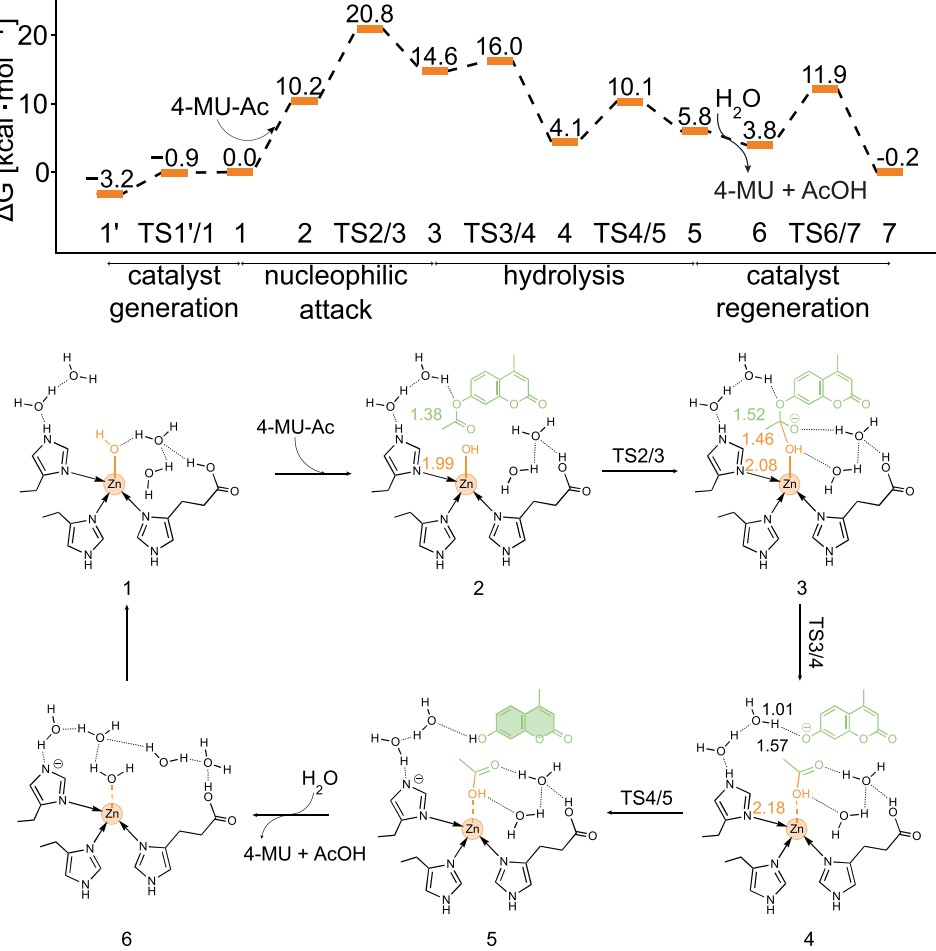

**Fig. 3 | Reaction mechanism of Zn²⁺-dependent pathway.** Calculated free energy profile for the proposed catalytic mechanism for 4-Mu-Ac hydrolysis at a Zn²⁺ catalytic center, along with the structures involved in the reaction profile. The length is in angstrom. Source data are provided as a Source Data file.

contribute to the inhibitory effect of Zn²⁺ on catalytic performance. Collectively, these results indicate that dense-phase basicity, internal mobility, and Zn accumulation within the condensates act in concert to regulate catalytic activity (Fig. 1c).

## Molecular level studies of Zn²⁺-mediated catalytic capacity of condensates

To further understand the mechanistic role of Zn²⁺, we constructed an atomic model of the condensate R2H/E2H and carried out classical molecular dynamics (MD) simulations. The pair distribution function of His $C_\alpha$ atoms calculated from trajectories showed a significant peak in the range of 6.5-7.5 Å (Supplementary Fig. 11). Based on this, we designed a catalytic center consisting of two peptides, a Zn²⁺ ion, neighboring water molecules, and the substrate, where two His residues from one peptide and one His residue from the other peptide coordinates the Zn²⁺. In this model, the distance between the $C_\alpha$ atoms of the two His residues is 6.67 Å, consistent with those obtained from the MD simulations. Then, we performed ab initio calculations to obtain a stepwise reaction mechanism.

As detailed in Fig. 3, the reaction proceeds in four steps: first, Zn²⁺ activates its bound water molecule, and a proton transfers from the water to the C-terminus to form a hydroxide group in complex 1, with a free energy 3.2 kcal/mol higher than complex 1′. Then, the substrate approaches, and the hydroxide ion nucleophile attacks the carbon atom of the substrate's ester bond, with a free energy barrier of 20.8 kcal/mol. This interaction initiates the formation of a new C-O (hydroxyl) bond (from 2.73 Å in compound 2 to 1.46 Å in compound 3)

through transition state TS2/3. Concurrently, the substrate's ester bond C-O distance extends from 1.38 Å to 1.52 Å in compound 3, indicating a propensity for cleavage.

Intermediate 3 is followed by breaking the ester bond with a free energy barrier of 16.0 kcal/mol (TS3/4), forming acetic acid and 4-methylumbelliferone phenolate ion in complex 4. The latter can easily take a proton from the uncoordinated nitrogen of His, mediated by two water molecules, ultimately forming the fluorescent alcohol 4-MU. The free energy of the resulting complex 5 is 4.4 kcal/mol lower than that of complex 2. In the last step, the product dissociates from the catalytic center, and a new water molecule enters the catalytic center (complex 6), with a free energy change of -2.0 kcal/mol. Following this, a proton from the Zn²⁺ bound water molecule transfers to the deprotonated His via mediation by water molecules, regenerating the catalytic center structure as in complex 1 and initiating the next cycle.

The free energy barrier of 20.8 kcal/mol for the rate-determining step is fairly comparable with free energy barriers reported for pNPA hydrolysis by catalytic self-assembled structures[13,23], and lower than others[3], suggesting that the catalytic reaction can occur spontaneously at room temperature.

To confirm that the peptide directly interacts with Zn²⁺ ions, we performed CD analysis in the absence or presence of ZnCl₂. The CD analysis (Supplementary Fig. 12) shows that the His-containing peptides R2H and E2H retain a random-coil signature with a ~196 nm minimum and a ~225 nm maximum yet show distinct spectral shifts upon addition of ZnCl₂. For E2H, a new maximum near 203 nm appears

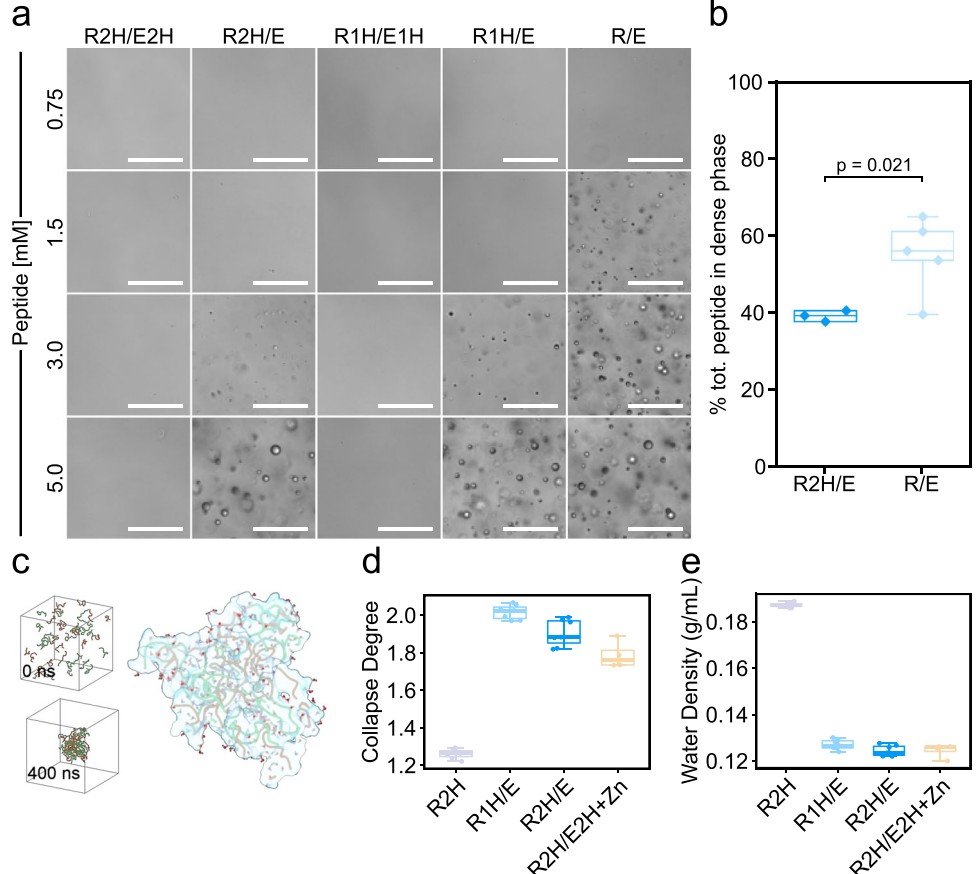

**Fig. 4 | Condensate formation through Zn$^{2+}$-free LLPS. a** Bright field microscopy images of condensates formed by LLPS at different concentrations of oppositely charged peptides (at 1:1 ratio) by different peptides combinations, in 10 mM Tris-HCl buffer at pH 7.5, scale bars=50 μm. The experiment was independently repeated twice. **b** Percentage of total peptide partitioned into the dense phase for the R2H/E and R/E systems, measured under the same experimental conditions as in panel (**a**). **c** Snapshots of the structures extracted from the trajectory of MD simulations of the R2H/E system (left panel), showing the initial structure and the structure after 400 ns of simulation, respectively, and a cross-sectional view of the R2H/E condensate displaying the distribution of water molecules inside (right panel). R2H and E are represented in red and green, respectively. **d** The collapse degrees of different systems, indicating that R2H is more soluble in water. **e** The water content in the condensate phase of different systems. d-e: [R2H] = 16 mM, [E] = 16 mM, [Zn$^{2+}$] = 150 mM (for R2H/E2H + Zn$^{2+}$). R2H, R1H/E, and R2H/E $n$ = 7 independent trajectories; R2H/E2H + Zn$^{2+}$ $n$ = 4 independent trajectories. Box hinges indicate the first and third quartiles, mid lines represent medians, and whiskers span the range of the data. Individual data points are shown. See Supplementary Table 11 for details on statistical analyses. Source data are provided as a Source Data file.

and intensifies with increasing Zn$^{2+}$ concentration, while R2H displays a progressive decrease in its random-coil minimum under the same conditions. In contrast, peptides lacking His show no detectable spectral changes in the presence of Zn$^{2+}$, indicating that the metal interacts specifically with His-containing peptides. In addition, we leveraged the intrinsic fluorescence of Trp within the peptides to perform Trp emission analysis in which environmental changes to the indole side chain results in changes in the emission intensity ($\lambda_{ex}$ = 280 nm). The results show a significant increase in the Trp emission intensity for the His-containing peptides, R2H and E2H, in the presence of 0.67 mM ZnCl$_2$ while no change is observed for the peptides lacking His, R and E (Supplementary Fig. 12).

Together, the CD and Trp emission analysis show that Zn$^{2+}$ ions interact with the His-containing peptides. These observations strengthen the phase diagrams and FRAP results which show that Zn$^{2+}$ promotes phase separation and slows the dense phase internal mobility, likely by strong interactions with R2H and E2H.

## Zn$^{2+}$-free formation of biomolecular condensates

In light of the inhibitory effect of Zn$^{2+}$ on condensate catalytic capacity, we next examined catalysis in the absence of ZnCl$_2$ as an alternative mechanistic regime. We first analyzed the LLPS propensity of

oppositely charged peptides (at 1:1 ratio) in the absence of ZnCl$_2$. Structural characterization revealed that without Zn$^{2+}$, no condensates are formed by R2H/E2H and R1H/E1H within the same concentration range used in the Zn$^{2+}$-mediated system (Fig. 4a), even after increasing the peptide concentration tenfold. Subsequently, we analyzed the LLPS propensity of various combinations of the positively charged His-containing peptide (R2H, R1H) with the negatively charged peptide (E). LLPS was observed for all combinations with E, at a concentration of 3 mM (Fig. 4a). In contrast to the trend observed in the Zn$^{2+}$-dependent system, the R/E system exhibits the highest LLPS propensity (Fig. 4a). Similarly, the dense-phase peptide concentration of R/E is 1.5-fold higher than that of R2H/E (Fig. 4b) further showing the higher efficiency of phase separation in this system. This suggests that the Zn$^{2+}$-mediated and the Zn$^{2+}$-free LLPS are driven by different interactions, with the former likely facilitated by interactions between His or Glu residues and the Zn$^{2+}$ ions[56]. These observations indicate that LLPS is mediated by a complex interplay of intermolecular attractive forces rather than solely electrostatic interactions, where Tyr plays a distinctive role in promoting LLPS[36].

MD simulations of Zn$^{2+}$-free LLPS systems show that the R2H/E system gradually aggregates from the initial random distribution to form a condensed, water-containing phase (Fig. 4c). We measured the

aggregation propensity of different peptides by calculating the degree of collapse, defined as the ratio of the solvent-accessible surface area (SASA) of the fully random state in the initial state to that during the simulation, where higher collapse degree indicates tighter peptide packing. As shown in Supplementary Fig. 13, in molecular dynamics simulations, the formation of peptide condensates reached equilibrium after ~200 ns. The R2H soluble peptide, which does not phase separate by itself, shows a lower collapse degree than R2H/E (Fig. 4d), indicating its higher water solubility. We defined the solvent-accessible surface of the peptide condensate as the boundary and calculated the enclosed volume and the number of water molecules within this boundary to determine the water content in the condensate during the simulation. The water density of the different systems is consistent with their collapse degree (Fig. 4e). Time-lapsed microscopy analysis showed that the R2H/E and R/E condensates coalesce and fuse over time, confirming their nature as hydrated liquid assemblies (Supplementary Videos 4-5).

## Zn$^{2+}$-free catalytic performance of peptide condensates

We determined product formation (Supplementary Fig. 14) and $V_0$ for each system as described above by using specific calibration curves of 4-MU (Supplementary Fig. 15), focusing on the lowest possible concentration of the peptides which undergo LLPS (3 mM) without ZnCl$_2$. In the absence of Zn$^{2+}$ ions, the His-containing condensates demonstrate a remarkable ability to enhance catalysis as a function of His content (Fig. 5a). At 0.7 mM 4-MU-Ac the $V_0$ of R2H/E is 4.2-fold higher than that of R2H/E2H with 0.20 mM ZnCl$_2$ demonstrating the Zn$^{2+}$-free condensates superiority over the Zn$^{2+}$-dependent condensates (Fig. 1). Therefore, we calculated the kinetic parameters for these systems (Supplementary Table 7). Among the three systems tested (R2H/E, R1H/E and R/E), the R/E system shows minimal increase in $V_0$ values with increasing substrate concentration (Fig. 5a) and exhibits the lowest $V_{max}$ and $k_{cat}$ values (Supplementary Table 7). In contrast, the $k_{cat}$ value of R2H/E and R1H/E is 3.5 and 7.0-fold higher than that of R/E. At 0.7 mM 4-MU-Ac, the $V_0$ of the R2H/E condensates is 1.3-fold and 6.9-fold higher than that of R1H/E and R/E, respectively (Fig. 5a, and Supplementary Table 8). After 2.5 min R/E showed <2% conversion, highlighting the critical role of His in catalysis. Moreover, when monitored for 60 min, R2H/E and R1H/E reach a plateau of fluorescent signal, while R/E is still in the linear phase (Supplementary Fig. 16).

To dissect the determinants of condensate activity, we tested whether replacing Arg with Lys would alter LLPS propensity. Strikingly, K2H/E did not undergo LLPS even at high peptide concentrations (10 mM; Supplementary Fig. 17), highlighting the critical role of Arg-mediated interactions in driving phase separation[36], whereas His-Tyr interactions alone are insufficient. CLSM imaging showed progressive accumulation of 4-MU fluorescence within R2H/E condensates over the course of the reaction (Fig. 5b). To analyze partitioning, 4-MU was first dissolved in MeCN and then added to preformed R2H/E droplets such that the final MeCN concentration was 2.8%. Under these conditions the fluorescence intensity of 4-MU in the dense phase was 24.3-fold higher than in the dilute phase (Fig. 5c).

To elucidate the catalytic contribution of LLPS, we conducted control kinetic experiments with 0.7 mM 4-MU-Ac. The $V_0$ of the R2H/E and R1H/E condensates were 81- and 63-fold higher, respectively, than spontaneous ester hydrolysis in buffer (Supplementary Table 8). Supernatants collected from centrifuged LLPS samples displayed reduced activity (1.8- and 1.7-fold lower than the two-phase systems), (Supplementary Table 8), and contained small heterogeneous microscopic structures, suggesting that residual assemblies may contribute to catalysis. Next, we compared condensates with each of the peptides alone, as a control. At the same peptide concentration (3 mM), individual R2H or E peptides—unable to undergo LLPS—displayed only slightly reduced $V_0$ compared to the condensates (Supplementary Table 8). Moreover, R2H and R1H peptides showed markedly higher $V_0$

values compared to the peptide R which lacks His (16.9- and 14.8-fold above R, respectively), confirming the essential role of His in promoting hydrolysis. Yet, this effect was not attributable to free His alone, since 3 mM His displayed far lower catalytic activity—5.0- and 4.4-fold lower than R2H and R1H, respectively (Supplementary Table 8). To probe the structural basis of this activity, we used DLS and transmission electron microscopy (TEM) to test whether the R2H and R1H alone form nanoscale assemblies. Indeed, R2H and R1H produced nanostructures averaging 281.3 ± 92.4 nm and 395.1 ± 154. 4 nm in diameter, respectively (Fig. 5d). Addition of E triggered LLPS, resulting in micron-sized droplets with an average diameter of 1.1 ± 0.43 μm for R2H/E, 1.3 ± 0.42 μm for R1H/E, and 2.2 ± 0.30 μm for R/E (Fig. 5d). TEM confirmed the formation of amorphous nanometric assemblies by R2H (Supplementary Fig. 18), consistent with previous reports that such supramolecular structures enhance hydrolysis1[10,11]. Despite their catalytic activity, the soluble peptides displayed a propensity to form low-abundance crystals. Bright-field imaging of R2H revealed elongated crystals emerging within minutes of substrate addition (0.7 mM, t = 3 min; Supplementary Fig. 19a). In contrast, condensates suppressed crystal formation, even after prolonged reaction times (15 min; Supplementary Fig. 19b), suggesting that sequestration of substrate and/or product within the dense phase prevents crystallization.

To shed light on the catalytic capacity of the condensates, we measured the pH of the dense and dilute phases using the ratiometric probe SNARF-1. Both phases in the R2H/E and R/E systems were found to be basic, with dilute-phase pH values of 8.0 ± 0.1 and 8.4 ± 0.3, respectively (Fig. 5e, f, and Supplementary Table 6). The basic environment in both phases is expected to promote the hydrolytic reaction, consistent with the enhanced catalytic activity observed in bulk buffer under basic conditions (Supplementary Fig. 16). FRAP analysis of R2H/E and R/E condensates (Fig. 5g) suggested that both condensates are liquid assemblies (Fig. 5h) that exhibit typical coalescence (Supplementary Videos 4-5), with almost full recovery after photobleaching (95.7 ± 2.6% and 99.9 ± 2.6% for R/E and R2H/E, respectively). Yet, R2H/E condensates had 4.7-fold faster $t_{1/2}$ value than that of R/E condensates (1.4 ± 0.3 sec for R2H/E and $t_{1/2}$ = 6.5 ± 1.5 sec for R/E) (Fig. 5i).

Together, these results suggest that the enhancement in ester hydrolysis is amplified by phase separation through substrate partitioning, elevated local concentrations, and suppression of inhibitory crystallization. Tyr-driven LLPS is essential for forming dynamic, liquid-like condensates, while His content governs catalytic efficiency. Thus, optimal catalysis emerges from the interplay between peptide sequence, supramolecular assembly, and condensate microenvironment.

## The catalytic performance of Zn$^{2+}$-free condensates is mediated by LBHB

MD simulations indicated that intermolecular hydrogen bonding between His residues is probable within the condensed phase (Fig. 5j, k). A detailed analysis of the C$_\alpha$ atom distance distributions between His residues from distinct peptides shows that, within the interaction range where His side chains can interact (C$_\alpha$ atom distances of ~8 Å), R2H/E exhibits a higher frequency than R1H/E (Fig. 5j, k).

Building on these findings, we leveraged structural ensembles from MD, to construct a cluster model for the Zn$^{2+}$-free system and performed ab initio calculations (Fig. 6). This analysis revealed a catalytic pathway that parallels the Zn$^{2+}$-dependent mechanism but replaces the metal ion with His side chains as Lewis acids. Thus, His sidechain activate water molecules and stabilize the generated hydroxide ions through unique intermolecular low-barrier hydrogen bonds (LBHB) between two His residues on neighboring chains[57].

Our calculations revealed that His residues are distributed throughout the condensate interior, suggesting the primary catalytic driver is an internal hydrogen-bonding network rather than a surface-

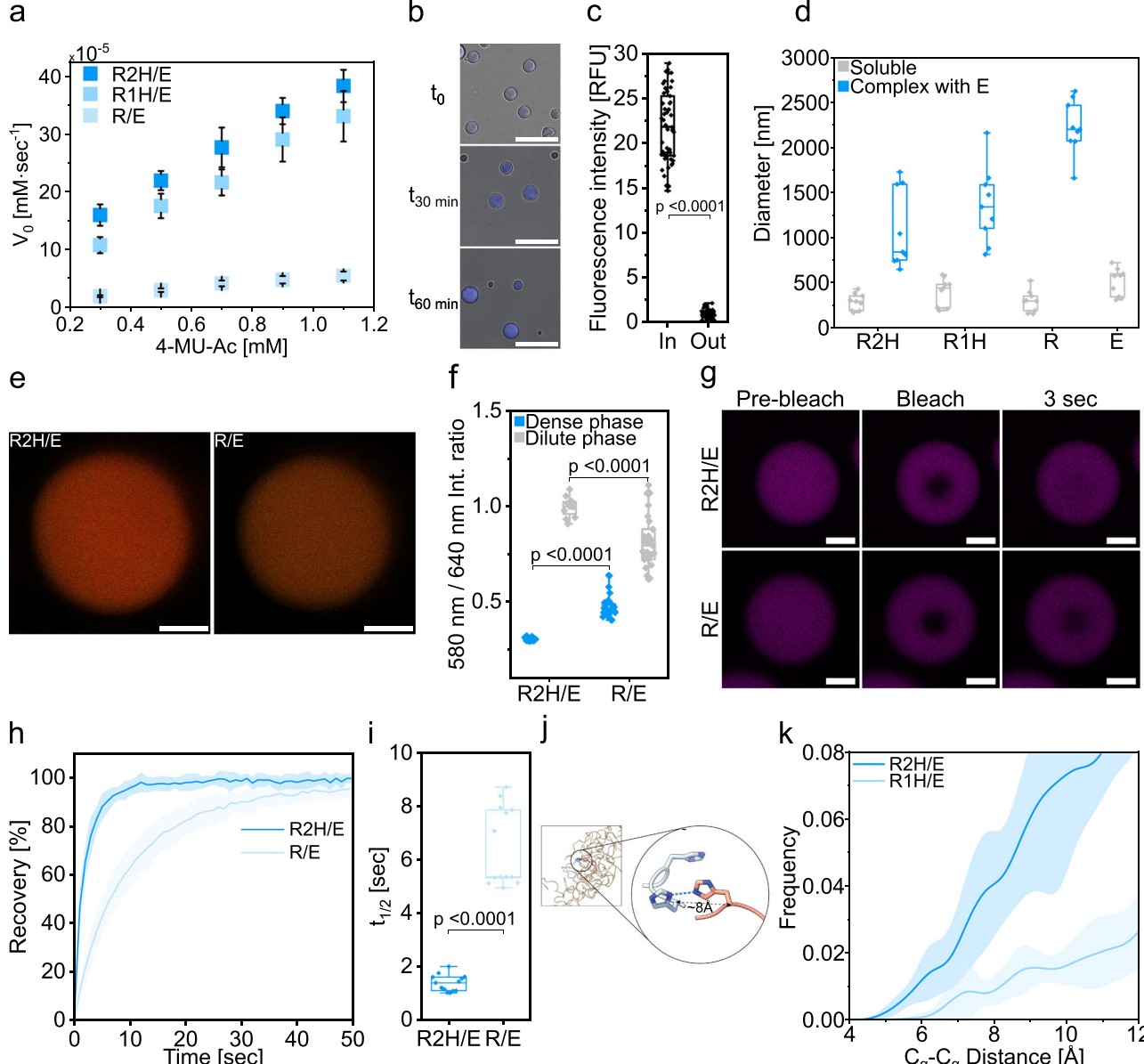

**Fig. 5 | Kinetic activity and physical properties of Zn²⁺-free condensates.**
**a** Kinetics of 4-MU-Ac hydrolysis catalyzed by R2H/E, R1H/E and R/E condensates (3 mM) in 10 mM Tris buffer at pH 7.5. Values represent averages, error bars represent S.D. **b** CLSM images of 4-MU-Ac (1.1 mM) ester hydrolysis in R2H/E condensates at t = 0, 30, and 60 min. The fluorescence of 4-MU product was monitored in the condensates by $\lambda_{ex}$ = 405 nm. Scale bars=15 μm. The experiment was independently repeated three times. **c** Fluorescence intensity of 4-MU in the dense vs. dilute phase of R2H/E condensates analyzed by CLSM. Individual data points are shown. Two-sided *p* values: *p* < 0.0001. **d** DLS analysis of nanoscale and microscale assemblies formed by soluble peptides (grey) or in combination with E, resulting in LLPS (blue). Peptide concentration is 3 mM in 10 mM Tris buffer at pH=7.50. Individual data points are shown. **e**–**f** CLSM analysis of the pH in dense phase of R2H/E and R/E condensates (Tris buffer, pH 7.5) obtained using SNARF-1 (20 μM).
**e** Representative CLSM images, scale bars=2 μm. The experiment was performed once under the conditions described. n numbers are indicated in Supplementary Table 6. **f** Box plot of the SNARF-1 580/640 nm emission intensity ratio measured in the dense and dilute phases of R2H/E and R/E condensates. The experiment was performed once under the conditions described. n numbers are indicated in

Supplementary Table 6. Individual data points are shown. Two-sided *p* values for dilute phase and condensates.: *p* < 0.0001. **g**–**i** FRAP CLSM analysis of R/E and R2H/E condensates obtained using rhodamine b as a fluorescent probe ($\lambda_{ex}$ = 560 nm). **g** CLSM images of droplets before, immediately after and 3 sec after photobleaching. Scale bars=2 μm. The experiment was independently repeated twice. **h** FRAP recovery plots, shaded areas represent ± S.D. **i** calculated $t_{1/2}$ values. Individual data points are shown, each represent one condensate from 2 independent measurements. Two-sided *p* values: *p* < 0.0001. **j** Snapshot of the MD trajectory of the R2H/E system. The enlarged panel shows the LBHB between a pair of His residues. The two R2H peptide molecules are colored red and blue, respectively. **k** The distribution of the distances between the $C_\alpha$ atoms of His residues. The shaded error bands correspond to the S.D. computed across independent MD trajectories. R2H/E exhibits a higher distribution than R1H/E, indicating that R2H/E has a greater likelihood to form LBHB. Box hinges indicate the first and third quartiles of the corresponding data, mid lines represent medians, and whiskers span the range of the data. See Supplementary Table 11 for details on statistical analyses. Source data are provided as a Source Data file.

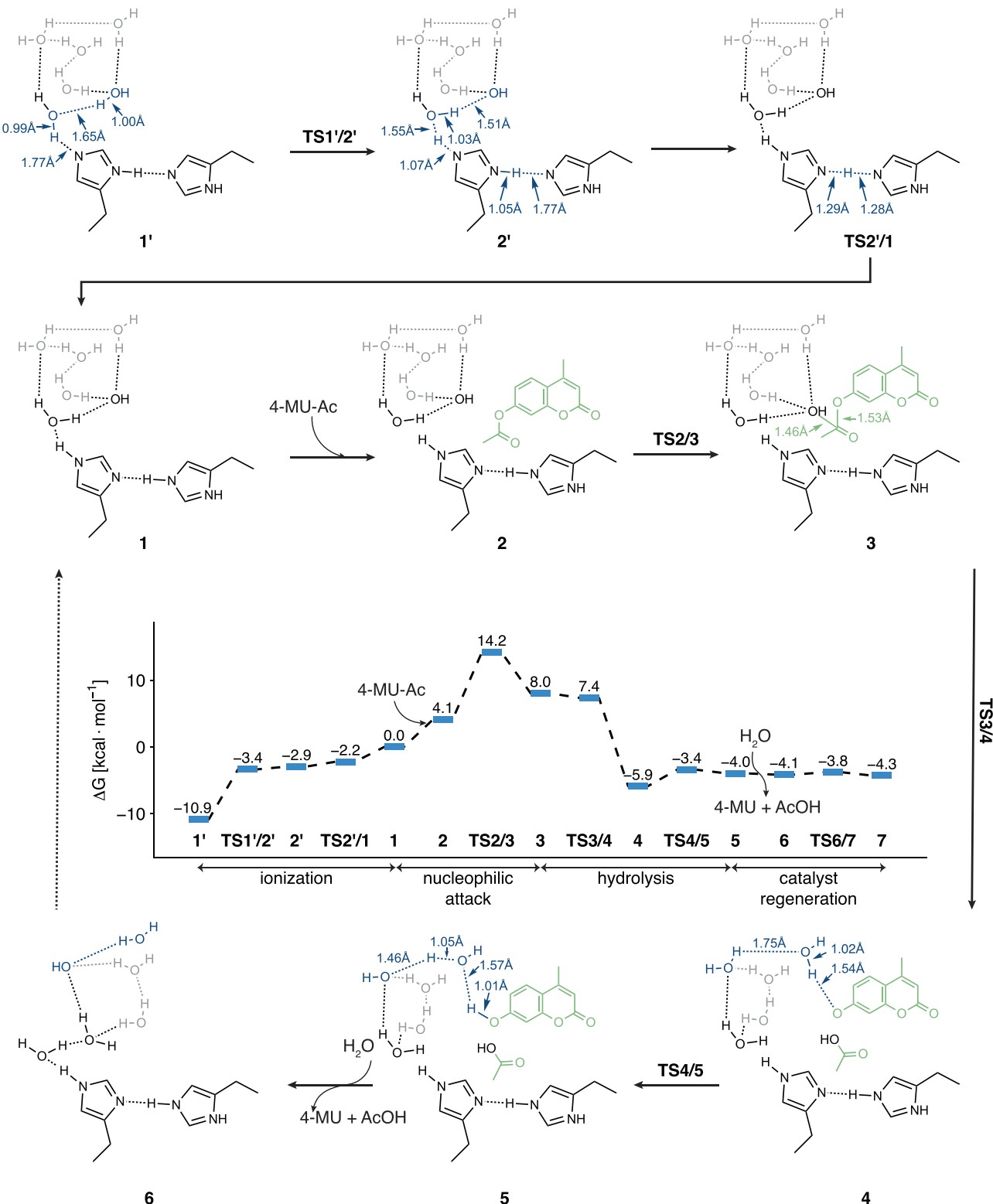

**Fig. 6 | Zn²⁺-free mechanism of catalytic condensates.** The free energy profiles for 4-Mu-Ac hydrolysis under metal-free conditions, along with the structures involved in the reaction profile. The key bonding processes and interactions in the model are highlighted with the interatomic distances (in Å) indicated. Source data are provided as a Source Data file.

specific mechanism (Supplementary Fig. 20–21). Thus, two adjacent His side chains can form near coplanar hydrogen bonds, both protonated at the $N_\delta$ position (Fig. 6, complex 1'), with a constrained $C_\alpha$-$C_\alpha$ distance of 7.86 Å. Importantly, a range of $C_\alpha$ distances (5-11 Å) can support LBHB formation by different $N_\delta/N_\varepsilon$, configurations on the

imidazole ring (Supplementary Fig. 22). In the first step, a neighboring water molecule donates a proton to one His via a water bridge, leading to formation of a hydroxide group and a positively charged His (complex 2'), with a free energy change of +8.0 kcal/mol. Consequently, the two His then form a LBHB in the "imidazolium-imidazole"

configuration, as characterized by the markedly short distance (~2.58 Å) between the donor and acceptor nitrogen atoms, and the negligible free energy barrier of only 0.7 kcal/mol (TS2'/1). The transfer of the shared proton further stabilizes the hydroxide ion.

Once stabilized, the hydroxide ion attacks the ester carbon of the substrate, initiating bond cleavage with a free energy barrier of 14.2 kcal/mol (TS2/3). Upon the cleavage of the ester bond, acetic acid and phenol are produced (complex 4; free energy change of 13.9 kcal/mol), followed by proton transfer to generate the 4-MU. In the final step, rearrangement of the hydrogen bond network between water molecules (TS6/7) resets the system, preparing the system for another catalytic cycle. The rate-determining step is the nucleophilic attack on the ester bond, with a free energy barrier of 14.2 kcal/mol, suggesting that this process can occur at room temperature. Notably, this barrier is lower than in the $Zn^{2+}$-dependent system, providing a mechanistic explanation for the enhanced catalytic efficiency of the $Zn^{2+}$-free condensates (Figs. 2e, and 5a).

Lastly, we modulated the protonation states of His residues within the R2H/E system to varying degrees in MD simulations (Supplementary Table 9). Considering that the pKa for the complete deprotonation of His to form its anionic form is ~13, we only consider its neutral and cationic forms. We compared the collapse degree and the pairwise His' $C\alpha$-$C\alpha$ distance distribution across neutral and partially protonated systems (Supplementary Fig. 23). The results demonstrate that His state does not significantly alter the LLPS properties of the peptide droplet (degree of collapse). Furthermore, the $C\alpha$ distance distribution remained largely unaffected, suggesting that the structural prerequisites for LBHB formation are maintained. These results suggest that while the local acid-base environment within the droplet may differ from that of the bulk phase, our proposed molecular mechanism remains relatively robust to pH variations. This stability might be attributed to histidine partially acting as a buffer agent, effectively maintaining the internal structural integrity of the LLPS under varying solution conditions.

## Discussion

Recent studies increasingly explore the utility of condensates for nanotechnological and catalytic applications. Two notable examples highlight the catalytic advantages of dense-phase environments: Guo et al[47]. reported that resilin-like polypeptide condensates display enhanced initial rates for ester hydrolysis, attribute to interfacial electric fields in the dense phase that lower the free-energy barrier for the reaction intermediate. Similarly, Reis et al.[46] showed that condensates formed by a catalytic peptide stabilize a folded β-hairpin and drive high partitioning of phosphorylated substrates into the dense phase, substantially increasing catalytic efficiency for pNPP hydrolysis. These studies convincingly demonstrate that condensate microenvironments can accelerate chemical reactions relative to the dilute phase, but they do so in systems where the catalytic function is effectively provided by a folded domain or physicochemical properties of the condensate environment.

In contrast, the system presented here relies on minimalistic His-rich peptide in which phase separation and catalytic activity are encoded within the same molecular scaffold, enabling catalysis in the dense phase and in the dilute phase. Similarly, Gupta et al recently showed catalytic His-containing tetrapeptide which forms non-equilibrium condensates through dynamic covalent bond formation[58]. In our system, the sequence-encoded design yields a dual catalytic regime, metal-dependent and metal-free, that provides insight into how primary sequence, supramolecular assembly, and emergent phase behavior jointly regulate reactivity. In this context, the condensate is not simply a passive concentrator of an external catalyst, but an emergent catalytic material whose mechanism, efficiency, and physical properties can be tuned through peptide sequence or cofactor availability.

A key insight from this study is that catalytic output in condensates is affected by dense-phase dynamics. $Zn^{2+}$ concentration simultaneously modulates peptide coordination state, local basicity, and internal molecular mobility. The links between coordination chemistry, microenvironmental pH, and diffusivity provides a mechanistic explanation for the counterintuitive observation that maximal phase separation does not coincide with maximal catalytic efficiency.

Comparing with previously reported self-assembling peptide catalysts for pNPA hydrolysis show that some ordered supramolecular systems achieve higher apparent catalytic than the R2H/E condensates for 4-MU-Ac hydrolysis[13]. However, direct comparisons are complicated by differences in substrate structure, solubility and intrinsic reactivity, and to our knowledge there are no comprehensive cross-platform studies that benchmark 4-MU-Ac hydrolysis across enzyme-mimetic catalysts. Thus, apparent differences in efficiency likely reflect substrate-dependent effects as much as intrinsic limits of the catalytic motif.

In the absence of $Zn^{2+}$, the condensates access a more efficient catalytic regime mediated by intermolecular LBHB between His residues. The lower free-energy barrier calculated for this metal-free pathway provides a mechanistic basis for the enhanced catalytic performance of $Zn^{2+}$-free condensates and highlights the catalytic potential of collective H-bonding networks in liquid materials.

This work opens avenues for employing condensates in diverse applications. The partitioning of the hydrophobic and poorly water-soluble product 4-MU into the dense phase highlights opportunities for localized synthesis and delivery. Notably, 4-MU has been widely studied as a chemopreventive agent[59] in pancreatic, prostate, breast, and colorectal cancers due to its inhibition of hyaluronan synthesis. Because the peptide condensates both generate and enrich 4-MU, they suggest strategies for localized production and accumulation of bioactive small molecules, as well as potential applications in environmental remediation, e.g., ester pollutant removal or polyester biodegradation, and the design of next-generation biocatalysts with enhanced selectivity and functionality.

## Methods

### Materials

Peptides were custom synthesized, then purified by high performance liquid chromatography to 95% and supplied as lyophilized powders by Genscript, Hong Kong. Unless otherwise specified, all reagents were of the highest available purity. 4-Methylumbelliferyl acetate was purchased from Tzamal D-Chem. 4-Methylumbelliferone was purchased from Rhenium. 5(and 6)-Carboxy SNARF 1 was purchased from LabSuit Shop. Acetonitrile, NaCl, NaOH and HCl were purchased from BioLab. $ZnCl_2$, Sigmacote and Trizma base were purchased from Merck.

### Phase diagrams and turbidity measurements

**$Zn^{2+}$-mediated phase diagram.** Stock solutions of positively charged peptide (solution A) and negatively charged peptide (solution B) were prepared at 1.5 mM in a 0.67 mM $ZnCl_2$ solution in 10 mM Tris-HCl and the pH of each solution was adjusted to 7.50 ± 0.04. Samples were formed by adding buffer, solution A and solution B, at final concentrations of 0.05, 0.01, 0.25, 0.5 mM of each peptide. For R/E system the following concentrations combinations were not evaluated due to the results obtained for R1H/E1H at the same conditions: 0.01/0.01, 0.01/0.05, 0.01/0.25, 0.01/0.5, 0.05/0.01, 0.25/0.01, 0.5/0.01.

**Turbidity measurements.** The turbidity of 36 µl of each condition was measured in quadruplicates in room temperature at λ = 350 nm using a Synergy H1 or Epoch2 microplate reader. For each condition, buffer background was subtracted from the average absorbance. Values below 0 were defined as 0. All experiments were performed in two

independent measurements. Data represents the corresponding intensity for the displayed images.

**Zn$^{2+}$-free phase diagram.** Stock solutions of solution A and solution B were prepared at 12 mM in 10 mM Tris-HCl and the pH of each solution was adjusted to 7.50 ± 0.04. Samples were formed by adding buffer, solution A, solution B and MeCN (final concentration of 2.8%), at final concentrations of 0.75, 1.5, 3, 5 mM of each peptide.

**Microscopy analysis of LLPS.** For both systems, the samples were imaged (50 µl) immediately after preparation in a 96-well black glass bottom plate, by fluorescence microscope (Olympus), x40/0.95 NA Plan Extended Apochromat objective. Images were collected and processed using CellSens Dimension software. Display ranges were adjusted individually for clarity of visualization.

## Effect of ZnCl$_2$ and NaCl on LLPS

All ZnCl$_2$-containing buffer solutions were prepared from an initial stock of 10, 7.68 or 1.67 mM ZnCl$_2$ in 10 mM Tris-HCl pH 7.50 ± 0.04, at final concentrations of 0.1, 0.2, 0.3 and 0.4 mM ZnCl$_2$. For CaCl$_2$ and NaCl analyses, peptides stock solutions were prepared in 0.67 mM salt in 10 mM Tris-HCl buffer. Stock solutions of R2H and E2H were prepared at 1.5 mM (final concentration of 0.5 mM of each peptide) in an appropriate buffer and the pH of each solution was adjusted to 7.50 ± 0.04. Samples were formed by adding buffer, R2H and E2H. The samples were imaged as described above.

## Total peptide content in the dense phase

**Calibration curves.** Stock peptides solution (positive and negative) at concentration of 1 mM were prepared in 10 mM Tris buffer. The peptide solutions were mixed and dilute with buffer and 2.8% ACN to achieve final peptide concentrations of 0.1, 0.08, 0.06, 0.04 and 0.02 mM from each peptide. 36 µl of each peptide solution was added to a 384 well plate. Absorbance spectra of quadruplicates measured at range of 230-300 nm in a 384 well black plate by Biotek H1 synergy plate reader (purchased from Lumitron, Israel). The calibration curves were calculated at 322 nm.

**Zn$^{2+}$-dependent.** Solutions of 1.5 mM E2H and R2H peptides were prepared in 10 mM Tris with 0.2, 0.33 and 0.67 mM ZnCl$_2$ at pH 7.5 ± 0.04. A solution of a final concentration of 0.5 mM of each peptide with 1 mM 2.8% ACN were made at volume of 144 µl in a 1.5 ml Eppendorf tube. After 10 min of incubation, the samples were centrifuged at 1,500 × g for 7 minutes. A volume of 120 µl from the supernatant was collected and diluted 4 times with 10 mM Tris buffer. The absorbance of triplicates of 36 µl of the diluted supernatant was measured at λ = 280 nm in a 384 well black plate by Biotek H1 synergy plate reader (purchased from Lumitron, Israel). All experiments were performed in triplicate. The concentration of the supernatant solutions determinate by the calibration curves.

**Zn$^{2+}$-free.** Solutions of 9 mM positive and negative peptides were prepared in 10 mM Tris at pH 7.5 ± 0.04. A solution of a final concentration of 3 mM of each peptide with 1 mM 2.8% ACN were made at volume of 144 µl in a 1.5 ml Eppendorf tube. After 10 min of incubation, the samples were centrifuged at 17,700 × g for 1 h at 10 °C. A volume of 120 µl from the supernatant was collected and diluted 4 times with 10 mM Tris buffer. The absorbance of triplicates of 36 µl of the diluted supernatant was measured at λ = 280 nm in a 384 well black plate by Biotek H1 synergy plate reader (purchased from Lumitron, Israel). All experiments were performed in triplicate. The concentration of the supernatant solutions determinate by the calibration curves.

%EE was calculated using Eq. 1.

$$\%EE = \frac{C_T - C_{\text{sup}}}{C_T} \tag{1}$$

## Reaction kinetics analysis

Choice of substrate and working concentration range: We selected 4-methylumbelliferyl acetate (4-MU-Ac) because the fluorescent 4-MU product allows simultaneous kinetic monitoring and spatial imaging by confocal laser scanning microscopy—measurements that are central to the mechanistic claims in this work. Absorbance-based substrates (e.g., pNPA/pNPP) are more soluble and permit broader bulk concentration ranges, but absorbance readouts in intact two-phase samples are strongly affected by light scattering from droplets, which evolves during LLPS and cannot be reliably blanked; therefore, absorbance assays were not used on intact condensates here. We empirically selected a working substrate window of 0.3–1.1 mM as concentrations above 1.1 mM promoted rapid crystal formation in several peptide sequences and reduced reproducibility.

Stock solution (39.6 mM) of 4-MU-Ac was prepared in MeCN, lower concentrations were diluted from this stock in MeCN (32.4, 25.2, 18 and 10.8 mM, for final concentrations of 1.1, 0.9, 0.7, 0.5 and 0.3 mM respectively). Stock solutions of solution A and solution B were prepared at 1.5 mM in 0.67/0.33/0.20 mM ZnCl$_2$ solution in 10 mM Tris-HCl and the pH of each solution was adjusted to 7.50 ± 0.04. Samples were prepared directly in wells. Substrate was added lastly from the appropriate stock by pipetting into the well, at a final concentration of 0.5 mM of each peptide. The fluorescence intensity was immediately measured ($\lambda_{ex}$ = 360 nm, $\lambda_{em}$ = 450 nm) for 3 min in a 384-well black plate with a clear bottom or a UV plate, by a Biotek H1 synergy plate reader. The $V_0$ was obtained by slope of data acquired up to 2.5 min using a linear fit and divided by the calibration curve slope. Soluble peptides were at 0.5 mM final concentration. The same protocol was used for Zn$^{2+}$-free condensates (R2H/E, R1H/E, R/E), with appropriate peptide concentrations (final concentration of 3 mM of each peptide). Zn$^{2+}$-free soluble peptides were at 3 mM final concentration. Kinetic parameters were calculated by fitting the data to the Lineweaver Burk plot using linear regression, the calculated errors were calculated using the errors obtained from the linear fit. Catalyst concentration was set as 3 mM for all systems. For the 1 h kinetics analysis, the measurements were performed for 60 min. For kinetics at pH 9 the pH of the buffer was adjusted to 9 ± 0.04. Control samples (addition of MeCN instead of substrate) were subtracted from the average fluorescence intensity value. Displayed results are one representative experiment from three independent measurements.

**Conversion calculations.** For each system, control samples (addition of MeCN instead of product) were subtracted from the average fluorescence intensity value at 2.5 min, and then divided by the corresponding calibration curve's slope. The conversion was calculated by Eq. 2:

$$\%\text{Conversion} = \frac{product\ concentration\ [mM]}{0.7\ [mM]} \cdot 100 \tag{2}$$

## Calibration curves

Calibration curves were prepared to each system separately.

Stock solution (7.2 mM) of 4-MU was prepared in MeCN, lower concentrations were diluted from this stock in MeCN (5.4, 2.7, 0.9, 0.225, 0.16, 0.08 and 0.04 mM for final concentrations of 0.2, 0.15, 0.075, 0.025, 0.006, 0.004, 0.002 and 0.001 mM, respectively). Peptide solutions were prepared in 10 mM Tris-HCl and the pH was adjusted to 7.50 ± 0.04. Condensate solutions were prepared directly in wells. 4-MU was added lastly from the appropriate stock by pipetting into the well. The fluorescence intensity was measured

($\lambda_{ex} = 360$ nm, $\lambda_{em} = 450$ nm) in a 384-well black plate with a clear bottom or UV plate by a Biotek H1 synergy plate reader. Control samples (addition of MeCN instead of product) were subtracted from the data. Calibration curves values obtained by averaging the data acquired in two independent experiments.

## pH measurement using SNARF-1

All solutions were imaged in coated 96-well plates with sigmacote, according to a previously reported protocol by Rosen and coworkers[60]. Briefly, plates were first cleaned by incubation in 2% cuvette cleaner solution (30 min), followed by extensive rinsing with tap water and DDW and air-drying. Then, wells were incubated with 1 or 10 M KOH (200 μL per well) for 1 h, rinsed thoroughly with tap water and DDW, and dried. Wells were then coated with Sigmacote (100 μL per well, immediately removed), air-dried for 10 min, washed twice with isopropanol, and dried in a chemical hood. Coated plates were sealed and stored at 4 °C until use. Before each experiment, the well was incubated for ~30 min with Pluronic F-127 solution (5 mg/mL in buffer without ZnCl₂), washed five times with buffer, and the sample was added immediately thereafter. Final peptide concentrations were 0.5 mM and 3 mM for the $Zn^{2+}$-dependent and $Zn^{2+}$-free systems, respectively. SNARF-1 was dissolved in DMSO to prepare a 10 mM stock solution, which was further diluted into 1.8 mM aliquots (18% DMSO in DDW) and stored at −20 °C. All samples contained SNARF-1 at a final concentration of 20 μM (0.2% DMSO) and MeCN at a final concentration of 2.8%. 50 μl were transferred to a coated plate and incubated. $Zn^{2+}$-dependent systems were incubated in the plate for 60 min prior to imaging, whereas $Zn^{2+}$-free systems were incubated for 30 min. Calibration curves were constructed using buffers spanning pH 6.5–9.0 in 0.5 pH unit intervals, prepared in 10 mM Tris−HCl containing 0/0.20/0.33/0.67 mM ZnCl₂. For calibration curves measurements, five different background ROIs were selected per well in a single imaging repeat. Confocal images were acquired using a laser scanning confocal microscope (Zeiss LSM 900 inverted microscope) equipped with a 63×/1.40 oil DIC M27 objective. SNARF-1 fluorescence was excited at 488 nm, and emission was collected in two spectral windows (570–617 nm and 630–700 nm). Z-stacks were acquired with a spacing of 0.25 μm. Images were collected and processed using Zen Blue software. Images were taken at 2048 × 2048 pixels. For visualization and image export of 0.67 mM ZnCl₂, the histogram was cut and the upper limit was adjusted to 150 (white).

**Image analysis and pH determination.** Fluorescence intensities were extracted from the focal plane corresponding to the maximal probe signal within the relevant compartment, based on Z-stack acquisition. For each ROI, fluorescence was collected from the two emission channels and ratiometric values were calculated. For calibration curves, the ROI size was set to the average ROI size used for condensate analysis within each system. Calibration curves were fitted using a logistic function. For condensate analysis, circular ROIs with diameters slightly smaller than the corresponding droplet size were used to avoid edge effects. ROI diameter ranges [μm] were 4.69–5.95, 5.17–6.92, 5.06–6.44, 5.73–6.42, and 5.09–6.93 for R2H/E2H + 0.20 mM ZnCl₂, R2H/E2H + 0.33 mM ZnCl₂, R2H/E2H + 0.67 mM ZnCl₂, R2H/E and R/E respectively. For analysis of the dilute phase, circular ROIs with a diameter of 3 μm were placed throughout background regions of the image, ensuring no overlap between ROIs and avoiding condensates in the current or adjacent focal planes. When condensates within the same image were present in different focal planes within the same field of view, background ROIs were selected independently in each corresponding plane and averaged. The number of background ROIs per image varied due to differences in condensate density. Fluorescence ratios from all background ROIs within the same image

were averaged to yield a single background value per image, which was then used for pH calculation. pH values were determined by solving the resulting equations using Wolfram Alpha.

**Spectroscopic measurements.** The fluorescence intensity was measured ($\lambda_{ex} = 488$ nm, $\lambda_{em} = 520$-700 nm) in a 384-well black plate with a clear bottom by a Biotek H1 synergy plate reader.

## FRAP CLSM analysis

FRAP experiments were performed using a Zeiss LSM 900 confocal microscope to track the fluorescence of 1% rhodamine b as payload from 10 μM stock (in DDW). Peptide solutions were dissolved in 10 mM Tris-HCl with 0/0.20/0.33/0.67 mM ZnCl₂ and the pH was adjusted to 7.50 ± 0.04, as described above with 0.5 mM and 3 mM final concentrations of each peptide for the $Zn^{2+}$-dependent and $Zn^{2+}$-free system, respectively. Condensate solutions were prepared in Eppendorf tubes with MeCN (final concentration of 2.8%), 50 μl were transferred to a coated plate and incubated for 30-60 min before the experiment. All solutions were imaged in Pluronic F-127−coated 96-well plates (5 mg/mL in buffer without ZnCl₂) according to a previously reported protocol by Rosen and coworkers[60], using 10 M KOH. Photobleaching was performed on a circular region of interest (ROI) with a diameter of 1 μm, with a reference ROI of the same size and a background reference ROI of 1 μm. Condensates were 5.6-6.8 μm in diameter for $Zn^{2+}$-dependent condensates and 5.6-6.5 μm in diameter for $Zn^{2+}$-free. Five iterations of the 488 nm and 561 nm excitation lasers at 100% intensity each were applied for rhodamine B using a 40×/1.2 Imm Korr DIC M27 objective. Images were taken at 256 × 256 pixels. Fluorescence recovery at the bleached area was recorded and analyzed using Zen Blue 3.2 software. Photobleaching correction and recovery times were calculated using OriginPro 2025, learning edition software. Intensity data were normalized between 0% (bleach intensity) and 100% (pre-bleach intensity) using the following equation for raw data correction:

$$I_{corrected}(t) = R_1(t) - R_3(t) \times \frac{R_2^{pre} - R_2^{pre}}{R_2(t) - R_3(t)} \quad (3)$$

$R_1(t)$ is the FRAP region at time t, $R_2(t)$ is the reference region at time t, $R_3(t)$ is the background refence, $R_2^{pre}$ is pre-bleach average for the refence area ($R_2$) and $R_3^{pre}$ is pre-bleach average for the background area ($R_3$).

The following Eq. (4) used for min-max scaling (0% to 100% normalization):

$$I_{normalized}(t) = \left( \frac{I_{corrected}(t) - I_{min}}{I_{pre} - I_{min}} \right) \times 100 \quad (4)$$

$I_{corrected}(t)$ is the calculated intensity from Eq. (2), $I_{min}$ is the minimum fluorescence intensity and $I_{pre}$ denotes the pre-bleach average of the fluorescence intensity.

Then, the normalized data was fitted to the next equation to extract the fitting parameters:

$y = a - bc^x$, when a is the recovery that was measured. $t_{1/2}$ values of were calculated for each droplet using a,b,c extracted from the fitting, with the following equation: $t_{1/2} = \log_c \left( \frac{0.5*a}{b} \right)$.

$t_{1/2}$ values were calculated for each condensate separately, at 180 sec for $Zn^{2+}$-dependent condensates and 50 sec for $Zn^{2+}$-free condensates. Recovery curves represent averages of recovery plots, at 5 sec intervals for $Zn^{2+}$-dependent condensates and at 1 sec intervals for $Zn^{2+}$-free condensates.

## CLSM microscopy imaging of coalescence

All solutions were imaged in Pluronic F-127−coated 96-well plates (5 mg/mL in buffer without ZnCl₂) as described above. Peptide solutions were dissolved in 10 mM Tris-HCl with 0/0.20/0.33/0.67 mM

ZnCl$_2$ and the pH was adjusted to 7.50 ± 0.04, as described above with 0.5 mM and 3 mM final concentrations of each peptide for the Zn$^{2+}$-dependent and Zn$^{2+}$-free systems, respectively. All samples were prepared directly in the well and contained MeCN at final concentration of 2.8%. Zn$^{2+}$-dependent systems were incubated for ~30 min in the plate and Zn$^{2+}$-free were imaged almost immediately after preparation. Images were taken using a laser scanning confocal microscope (Zeiss LSM 900 inverted microscope), 40×/1.2 Imm Korr DIC M27 objective. Images were collected and processed using Zen Blue software. For visualization and image export, the histogram upper limit was adjusted to 150 (white). Videos were exported at different frame rates to reduce file size, as detailed here: R2H/E2H + 0.20 mM ZnCl$_2$ 47.4 fps; R2H/E2H + 0.33 mM ZnCl$_2$ 15.80 fps; R2H/E2H + 0.67 mM ZnCl$_2$ 15.80 fps; R2H/E 31.60 fps; R/E 31.60 fps. The experiment was performed once.

### STEM-EDS

Samples were prepared as previously described. 2 µl of the sample solution were applied to a FCF400-Cu grid (FORMVAR/Carbon Film, 400 mesh copper) and incubated for 2 min. Excess solution was removed by blotting the grid with a filter paper, then the grid was left to air-dry for 10 min. Imaging and elemental mapping were performed on a probe-corrected Spectra 200 (S)TEM (Thermo Fisher Scientific) equipped with an X-type cold field-emission gun and a Super-X EDS system (Thermo Fisher Scientific). Operating conditions: 80 kV accelerating voltage, probe current ~95 pA, probe collection semi-angle ~30°, camera length 98 mm, column pressure ~5 × 10 − 6 Pa. Images were acquired using segmented bright-/dark-field detectors (Panther, 8 segments) and a HAADF detector (Fischione Instruments). Data were acquired and processed in Velox 3.15 (Thermo Fisher Scientific). For elemental mapping, zinc used as marker; line-scan intensities were normalized as $(I_x − I_{min})/(I_{max} − I_{min})$. Analysis performed on 3 condensates of each system and the data represent one condensate.

### EE analysis

**Calibration curves.** Stock solution of 4-MU (36 mM) was prepared in acetonitrile. It was diluted with acetonitrile to vary concentration (14.4, 11.52, 8.64, 5.76 and 2.88 mM). 1 µl of each 4-MU concentration was added to a 384 well plate with 35 µl of 10 mM Tris buffer solution to final 4-MU concentrations of 0.4, 0.32, 0.24, 0.16 and 0.08 mM, respectively. Absorbance spectra of quadruplicates measured at range of 300–400 nm in a 384 well black plate by Biotek H1 synergy plate reader (purchased from Lumitron, Israel). The calibration curves were calculated at 322 nm.

**Zn$^{2+}$-dependent systems using spectroscopic measurements.** solutions of 1.5 mM E2H and R2H peptides were prepared in 10 mM Tris with 0.2, 0.33 and 0.67 mM ZnCl$_2$ at pH 7.5 ± 0.04. A solution of a final concentration of 0.5 mM of each peptide with 1 mM of 4-MU were made at volume of 144 µl in a 1.5 ml Eppendorf tube. For control samples acetonitrile was added instead of 4-MU. After 10 min of incubation, the samples were centrifuged at 1500 × g for 7 minutes. A volume of 120 µl from the supernatant was collected and diluted 4 times with 10 mM Tris buffer. The absorbance of triplicates of 36 µl of the diluted supernatant was measured at λ = 322 nm in a 384 well black plate by Biotek H1 synergy plate reader (purchased from Lumitron, Israel). All experiments were performed in triplicate. The concentration of the supernatant solutions is determined by the calibration curves. %EE was calculated using Eq. 1.

**Zn$^{2+}$-free system using CLSM.** All solutions were imaged in Pluronic F-127–coated 96-well plates (5 mg/mL in buffer without ZnCl$_2$) as described for FRAP. Peptide stock solutions (9 mM) were dissolved in 10 mM Tris-HCl and the pH was adjusted to 7.50 ± 0.04. Condensate solutions were prepared in Eppendorf tubes with 0.2 mM (2.8% MeCN) final concentration of 4-MU and 3 mM of each peptide, 50 µl were transferred to a coated plate and incubated for 30 min before the experiment. z-stack imaging with spacing of 0.2 µm were taken. Images show the z-stacking middle section and were taken using a Zeiss LSM 900 CLSM, 40×/1.2 Imm Korr DIC M27 objective. Images were taken at 1024 × 1024 pixels. Fluorescence images were taken using 405 nm laser. Images were collected and analyzed using Zen Blue software. Condensates were 2.5–4.2 µm in diameter. Each experiment repeated itself twice.

### CLSM imaging of 4-MU-Ac hydrolysis

All solutions were imaged in Pluronic F-127–coated 96-well plates (5 mg/mL in buffer without ZnCl$_2$) as described for FRAP (without incubation of samples). Reactions were prepared as detailed above for reaction kinetics analysis, 4-MU-Ac at 1.1 mM final concentration was added directly in the well. Z-stack images were taken immediately after adding the substrate solution and every 30 minutes up to 60 minutes. z-stack imaging with spacing of 0.2 µm were taken in each time point and each experiment repeated itself three times. Images show the z-stacking middle section and were taken using a Zeiss LSM 900 CLSM, 40×/1.2 Imm Korr DIC M27 objective. Images were taken at 1024×1024 pixels. Fluorescence images were taken using 405 nm laser. Images were collected using Zen Blue software. Each experiment repeated itself three times.

### Crystallization analysis

Stock solution of 4-MU-Ac was prepared in MeCN, final substrate concentration was either 1.1 mM or 0.7 mM. Stock solutions of R2H and E were prepared at 9 mM in 10 mM Tris-HCl and the pH of each solution was adjusted to 7.50 ± 0.04. Samples were formed by adding buffer, R2H and E (for condensates) to a final concentration of 3 mM of each peptide. Substrate was added lastly from the appropriate stock by pipetting into the well. Bright field images were taken for three samples by fluorescence microscope (Olympus), x10/0.25 NA Plan Apochromat objective (for soluble R2H with 0.7 mM 4-MU-Ac) and by x40/0.95 NA Plan Extended Apochromat objective (for R2H/E condensates and soluble R2H with 1.1 mM 4-MU-Ac). All samples were loaded on a 96 well Black Glass bottom plate. Images were collected and processed using CellSens Dimension software.

**Soluble R2H with 0.7 mM 4-MU-Ac.** Images were collected over a time lapse of 15 min. Each well was scanned in a 3 by 3 array with spacing of 86 µm between the rows and columns and a z-stack with spacing of 20 µm at a total range of 200 µm. Each experiment repeated itself three times.

**R2H/E condensates and soluble R2H with 1.1 mM 4-MU-Ac.** Images were collected manually by scanning the well at different locations and z-stacks for 15 min. Each experiment repeated itself three times.

### Dilute phase kinetics

Peptide stock solutions were prepared at final concentration of 3 mM of each peptide in 10 mM Tris-HCl and the pH of each solution was adjusted to 7.50 ± 0.04. Condensate were prepared as described above. Samples were centrifuged at 17,700 RCF for 4 h at 10 °C. The supernatants were collected and imaged immediately as described above. Kinetics measurements were conducted as described above. Dilute phase kinetics were calculated using a calibration curve constructed in buffer under identical experimental conditions.

### Mass spectrometry analysis

Samples were prepared following the same protocol as the kinetic assays, with a final substrate concentration of 0.7 mM. After ~30 min incubation at room temperature, an equal volume of acetonitrile was added to disrupt the condensates, and samples were collected for further analysis. The analysis was performed once for representative

samples under each condition to confirm product identity. High-resolution mass spectrometry (HRMS) analyses were performed on a Waters G2 XS QTOF instrument using both electrospray ionization (ESI) and atmospheric pressure chemical ionization (APCI) in positive ion mode. Samples were analyzed by direct injection. The molecular ion corresponding to the protonated species $[M + H]^+$ was measured in APCI mode. The calculated exact mass for $[M + H]^+$ of $C_{10}H_9O_3$ is $m/z = 177.0552$. The experimentally determined masses were as follows: R2H/E2H + 0.20 mM $ZnCl_2$, $m/z = 177.0549$ ($\Delta$ = 0.3 mDa, 1.7 ppm); R2H/E2H + 0.33 mM $ZnCl_2$, $m/z$ 177.0553 ($\Delta$ = 0.1 mDa, 0.6 ppm); and R2H/E2H + 0.67 mM $ZnCl_2$, $m/z = 177.0552$ ($\Delta$ = 0.0 mDa, 0.0 ppm); R2H/E, $m/z = 177.0551$ ($\Delta$ = 0.1 mDa, 0.6 ppm); R1H/E, $m/z = 177.0547$ ($\Delta$ = 0.5 mDa, 2.8 ppm); R/E, $m/z = 177.0549$ ($\Delta$ = 0.3 mDa, 1.7 ppm).

## DLS analysis

*$Zn^{2+}$-mediated* condensates were prepared as follows: Stock solutions of positively charged peptide (solution A) and negatively charged peptide (solution B) were prepared at 1.5 mM in a 0.67 mM $ZnCl_2$ solution in 10 mM Tris-HCl and the pH of each solution was adjusted to 7.50 ± 0.04. Samples were formed by adding buffer, solution A, solution B, and MeCN (final concentration of 2.8%) at final concentration of 0.5 mM for R2H/E2H.

*$Zn^{2+}$-free* condensates were prepared as follows: Stock solutions of solution A and solution B were prepared at 9 mM in 10 mM Tris-HCl and the pH of each solution was adjusted to 7.50 ± 0.04. Samples were formed by adding buffer, solution A, solution B and MeCN (final concentration of 2.8%), at final concentrations of 3 mM for R/E, R1H/E and R2H/E.

Each sample was placed in ZEN0040 cuvettes and were measured by Malvern - Zetasizer nano Z (purchased from Dr. Golik). The diameter of the condensates was calculated as the average of 14 repeats. Each experiment was conducted in triplicates.

## CD

Peptide solutions for CD were prepared at concentration of 0.5 mM in 10 mM Tris buffer solution with 0.2, 0.33, 0.67 mM of $ZnCl_2$ and with no $ZnCl_2$ (in samples of mixtures of peptides, each peptide was at concentration of 0.5 mM). pH was adjusted to 7.5 ± 0.04. The solutions were placed in a 0.2 mm path length quartz cuvette at 25 °C, and the range of 190–260 nm was recorded on a Chirascan spectrometer. Background (buffer with suitable $ZnCl_2$ concentration) was subtracted from the CD spectra.

## Trp emission analysis

Samples of R, R2H, E and E2H peptides were prepared at final concentration of 0.5 mM with and without 0.67 mM $ZnCl_2$ in 10 mM Tris-HCl and the pH of each solution was adjusted to 7.50 ± 0.04. The fluorescence intensity was measured ($\lambda_{ex}$ = 280 nm, $\lambda_{em}$ = 300-500 nm) in a 384-well black plate with a clear bottom by a Biotek H1 synergy plate reader.

## Microscopy analysis of K2H/E, K/E

Stock solutions were prepared at 20 mM (final concentration of 10 mM of each peptide) in 10 mM Tris-HCl and the pH of each solution was adjusted to 7.50 ± 0.04. Samples were prepared by mixing K2H or K solution with E solution at 1:1 ratio. Samples were imaged as described above for the phase diagrams.

## TEM analysis

R2H sample was prepared at a final concentration of 3 mM, in 10 mM Tris-HCl buffer with 2.8% MeCN, pH 7.50 ± 0.04. An aliquot of 5 μL of the sample solution was applied to a FCF400-Cu grid (FORMVAR/Carbon Film, 400 mesh copper) and incubated for 2 min. Excess solution was removed by blotting the grid with a piece of filter paper, followed by staining with 5 μL of 50% (v/v) uranless for 1 min. After

blotting excess stain solution, the grid was left to air-dry. Buffer sample was prepared the same way. The negatively stained sample was imaged by a Talos F200i (S)TEM operated at 80 kV. Images were recorded using a Ceta-M 4 K × 4 K detector.

## MD simulations

The initial structures of the peptides used in the simulation (including R, R1H, R2H, E2H) were generated using ESMFold[61]. Subsequently, the structure of each single sequence was sampled using simulated annealing. Each polypeptide was solvated in a cubic aqueous box, ensuring a minimum distance of 15 Å between the peptide and the periodic boundary. The system then underwent energy minimization to remove potential steric clashes. An annealing step was then performed by heating the system to 2000 K for 5 ns, followed by a cooling and equilibration phase at 300 K for an additional 15 ns. To extensively sample the conformational landscape, 40 independent, parallel simulations were carried out for each polypeptide. To construct the multi-peptide systems, 20 peptide conformations, obtained from the preceding simulated annealing trajectories, were randomly placed within a cubic periodic box with dimensions of 12.7 × 12.7 × 12.7 nm³. This arrangement resulted in a concentration of ~16 mM for each polypeptide species. For simulation systems containing only the R2H sequence, $Cl^-$ ions were added to neutralize the charge. In systems designed to investigate the effects of $Zn^{2+}$, 150 mM $ZnCl_2$ was added. The entire system included ~68,000 water molecules and 210,000 atoms. Details of the simulated systems is listed in Supplementary Table 10.

The simulations were performed using the CHARMM36m force field[62] and the TIP3P water model[63]. All simulations were performed using GROMACS 2022[64]. Non-bonded van der Waals interactions were calculated using a cutoff of 1.2 nm, with a force-switching function applied between 1.0 and 1.2 nm to smoothly taper the potential to zero. Long-range electrostatic interactions were treated with the Particle Mesh Ewald (PME) method, employing a real-space cutoff of 1.2 nm. The system was energy minimized to remove bad contacts. Then the system was heated to 300 K at a constant volume using velocity-rescaling thermostat. Production simulations were then continued in the NPT ensemble at 1 atm pressure and 300 K for 400 ns. Temperature and pressure were controlled by the Nosé-Hoover thermostat[65,66] and the Parrinello-Rahman barostat[67], with coupling time constants of 1.0 ps and 5.0 ps, respectively. Bonds involving hydrogen atoms were constrained by the LINCS algorithm[68], and a time step of 2 fs was used. System trajectories were saved every 100 ps for subsequent analysis. The first 200 ns of the production simulation were treated as equilibration and discarded from subsequent statistical analyses. We carried out three replicate MD simulations for each system and obtained consistent results. To investigate the impact of pH variations, we modulated the protonation states of histidine residues within the R2H/E system to varying degrees. Snapshots at 400 ns was extracted from replicate simulations of the fully neutral system to serve as the initial structure for these new simulations. Subsequently, 1/3 and 2/3 of the R2H peptides were randomly selected, and one histidine residue on each selected chain was protonated. Following energy minimization, 100 ns simulations were conducted under identical simulation parameters. Data from the final 10 ns of the trajectories were used to analyze the system properties under different protonation states. UCSF ChimeraX[69] and VMD[70] were used to analyze the MD simulation trajectories.

**Ab initio calculations.** In the active site model of the system containing $Zn^{2+}$, three His residues participate in coordination. Two His residues retain their side chains and Cα atoms, while one His additionally retains the carboxylate group at its C-terminal backbone. The three His residues coordinate with $Zn^{2+}$ through their $N_\delta$ atoms. The $C_\alpha$ atoms of the His are frozen to mimic the constraints resulting from

interactions within the condensate. When comes to the $Zn^{2+}$-free catalytic center, structures sampled from MD simulations was used to build the initial structure, and the $C_\alpha$ atoms of the two His are also frozen to mimic the condensate environment.

All geometry optimizations and frequency analysis were performed using M06-2X[71]/def-TZVP[72] with the PCM solvation model[73]. For electronic energies calculations, the larger def2-TZVP basis set[74] was used along with the SMD solvation model[75]. For systems containing $Zn^{2+}$, LANL2DZ[76] was used to describe $Zn^{2+}$ in geometry optimizations, while SDD[77] was used in electronic energies calculation. Thermodynamic corrections from frequency analysis were incorporated into the solution-phase single-point energy to obtain Gibbs free energy, which were then used in the mechanistic discussion. All ab initio calculations were conducted using the Gaussian16 program[78].

### Statistics and reproducibility
For kinetics measurements, for each dataset, the mean of all measurements within a given condition was calculated, and the percentage deviation of each value from the mean was determined. All available measurements were included, even those from days with fewer replicates (minimum two replicates). To ensure consistency and accuracy in data analysis, exclusion criteria were determined for calibration curves and kinetic measurements based on the amount of data available and the range of raw data values. Outliers falling outside the range of more than 25% above or below the mean for kinetics data, or more than 30% above or below the mean for calibration curves, were excluded from the statistics. Separate cutoff ranges were selected to account for the differing variability in these datasets, with calibration curves exhibiting inherently higher variability. This approach ensured sufficient data retention to construct reliable calibration curves. Background measurements, which have very low fluorescence intensities and are inherently noisy, were visually inspected using graph representations, using only the first measurement. All calculations and exclusions were systematically documented to ensure reproducibility and transparency. Detailed experimental replicates are detailed in Supplementary Table 11. $p$ values are reported in the figure legends.

### Reporting summary
Further information on research design is available in the Nature Portfolio Reporting Summary linked to this article.

## Data availability
Unless otherwise stated, the data supporting the findings of this study are available within the paper and its Supplementary Information files. Source data are provided with this paper. Additional data are available at (https://doi.org/10.5281/zenodo.18770088). Source data are provided with this paper.

## Code availability
The inputs and outputs for the computational studies in this work are available within the supplementary materials. The computational studies in this work utilized the following third-party software packages:
Gaussian 16 (Rev. A.03)[78]: Commercial license available via (https://gaussian.com); NAMD 3.0[79]: Free for non-commercial use via (https://www.ks.uiuc.edu/Research/namd/); VMD 1.9.4[70]: Open-source visualization tool under academic license via (https://www.ks.uiuc.edu/Research/vmd/).

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

## Acknowledgements

A.L. acknowledges support from Israel Science Foundation (ISF) grant 3217/23. H.D. acknowledges the support from the National Natural Science Foundation of China (Nos. 22361142831, 22273034). A.L. acknowledges support from the European Research Council (ERC) grant CORE 101162920, and support from the Sagol Center for Regenerative Medicine. H.D. acknowledges the National Key R&D Program of China (No. 2023YB3813001), Jiangsu Province Front-end Technology Research and Development Program (BF2024056), the Frontiers Science Center for Critical Earth Material Cycling of Nanjing University. T.M. acknowledges support from the ADAMA Center for Novel Delivery Systems in Crop Protection for the PhD Fellowship and the Marian Gertner Institute for Medical Nanosystems for the Gertner Scholarship. T.M thanks Dr. G. Levi for the help with the TEM and STEM-EDS imaging. T.M thanks Dr. N. Tal for the help with the MS analyses. T.M thanks I. Katzir for assistance with FRAP experiments. Parts of the calculations were performed using computational resources on an IBM Blade cluster system from the High-Performance Computing Center (HPCC) of Nanjing University.

## Author contributions

A.L., A.B.L. and T.M. conceived and designed the experiments. H.D., Y.Y., and X.W. conceived and designed all computational analyses, including MD simulations and ab initio calculations. O.E and A.B.L helped analyzing the LLPS propensity and the crystallization analysis. Confocal microscopy analyses were conducted by A.B.L. and T.M. CD, Trp emission, DLS, STEM-EDS, peptide content and 4-MU partitioning in $Zn^{2+}$-dependent system were conducted by A.B.L. All other experimental analyses were conducted by T.M. T.M, A.L., Y.Y and H.D wrote the manuscript. All authors discussed and commented on the manuscript.

## Competing interests

The authors declare no competing interests.
