## [Transparent Peer Review File · Nature Communications]

Metal-dependent and metal-free mechanisms of peptide condensate catalysts

Corresponding Author: Professor Ayala Lampel

Version 1:

Reviewer comments:

Reviewer #1

(Remarks to the Author)

This paper demonstrated that: condensates formed by minimalistic histidine-containing peptides can catalyze ester hydrolysis via two pathways: a Zn^{2+} -dependent mechanism activating bound water and a metal-free mechanism driven by LBHB. The concept is of highly interested and the discovery is exciting. However, the authors want to address the following technical considerations before this can be published.

1. This work suffers from a fundamental flaw, namely the overlooking of the effect of pH on hydrolysis reactions. It is well known that pH affects hydrolysis reactions. Recently, more and more studies have shown that phase separation can create a distinct pH within condensates (see, doi.org/10.1101/2024.05.23.595321). The mechanism underlying this distinct pH environment is the partitioning of ions (including protons) during LLPS. In this study, the authors tested (1) condensates made of different peptides and (2) condensates formed in the presence of Zn^{2+} or other ions, both conditions that can create distinct intra-condensate pH. Therefore, the authors should first demonstrate that

(1) condensates made of different peptides possess the same (or at least similar) pH;

(2) condensates formed at different Zn^{2+} concentrations possess the same (or at least similar) pH.

Intra-condensate pH can be measured with fluorescent probes.

(3) study how distinct pH can modulate the reactions.

These experiments will provide alternative explanations to the current observations.

2. If, as is very likely, condensates made of different peptides possess different pH values, how do the authors explain the mechanisms underlying the observed hydrolytic activities?

3. The authors showed that condensates formed in the presence of Zn^{2+} become more compact, which in turn lowers their hydrolytic capability. However, more compact condensates may also be more hydrophobic inside, which would facilitate the partitioning of low-polarity molecules such as pNPA or 4-methylumbelliferyl acetate. Therefore, the authors should directly demonstrate the limited partitioning of 4-methylumbelliferyl acetate, for example by directly using 4-MU.

4. Using rhodamine B to probe partitioning and diffusion in this coacervate system can be problematic. Rhodamine B is a charged molecule, so its partitioning and diffusion can be affected by electrostatic interactions with the charged peptides. Because the hydrolysis substrates used in this article are non-charged, the authors should replace rhodamine B with a neutral fluorescent molecule.

5. The authors investigated the mechanism underlying the Zn^{2+} -independent hydrolytic activity of the coacervates and attributed it to LBHB formation. However, evidence for LBHBs within the Zn^{2+} -containing condensates is lacking. Since Zn^{2+} makes the condensates more compact, it could, in principle, promote LBHB formation and thus enhance the reaction — which would contradict the authors' conclusion that Zn^{2+} inhibits the hydrolytic capability of the coacervates. The authors should clarify this point.

6. For experimental rigor, please provide mass spectrometry data verifying the occurrence of hydrolysis reactions.

Reviewer #2

(Remarks to the Author)

The revised manuscript by Massarano et al. has been significantly improved compared to the original version in terms of both novelty and experimental rigor. First, the authors have reinforced the discussion part to clearly address the conceptual novelty that their work has demonstrated, including intrinsic catalytic properties and Zn-dependent switching between two distinct mechanisms of catalysis. (which is now also conveyed in the revised title). Additionally, the authors have included significant amount of rigorous kinetic studies, in addition to the FRAP and DLS experiments that were originally missing, to demonstrate the sequence-based control over the condensate catalytic and material characteristics.

Therefore, the authors have answered all of my points and concerns provided in the first round of review. I believe that the manuscript is now ready for the publication, and thus recommend it to be accepted. I believe that this work will provide a novel addition to the research field of LLPS, and improve our understanding of how condensate properties could be precisely tuned in LLPS-based systems.

Reviewer #4

(Remarks to the Author)

The manuscript by Lampel and co-workers describes the formation of LLPS peptide condensates capable of catalysing ester hydrolysis through either Zn²⁺-dependent or Zn²⁺-independent mechanisms. The authors combine microscopy, kinetic measurements, and computational studies to explore how histidine residues and metal coordination contribute to catalysis. Overall, the manuscript lacks a clear hypothesis-driven design, quantitative experimental validation of the proposed mechanisms, and the level of mechanistic generality expected for the journal.

Considering the level of novelty of the topic and the scope of the study, I do not believe that the manuscript is suitable for publication in Nature Communications, which addresses a broader multidisciplinary audience. A more specialised chemistry journal focusing on peptide-based catalysis or supramolecular systems would likely be a more appropriate venue.

- The manuscript focuses on peptide LLPS systems and their catalytic activity toward ester hydrolysis. Although this topic remains of chemical interest, it does not represent a particularly timely or conceptually novel advance for the Nature portfolio. Several studies have already reported catalytic coacervates and peptide condensates employing similar model reactions. The present work refines rather than expands the existing framework and therefore does not meet the broad conceptual impact expected for this journal.
- The rationale behind the experimental sequence requires clearer justification. The authors initially state that Zn²⁺ promotes LLPS and contributes to catalysis, but the data indicate that higher Zn²⁺ concentrations inhibit activity. This outcome is intriguing yet undermines the central design logic. It remains unclear why the system was optimised around Zn²⁺ if Zn²⁺ coordination ultimately suppresses reactivity. The authors should explicitly address whether the Zn²⁺-free system was conceived as a control or as an alternative catalytic regime.
- Some figure references are inconsistent or incomplete (e.g., “Fig. 1b–g” does not correspond to any figure panels). Furthermore, key datasets are represented only as heat maps or schematic diagrams, with no access to the raw numerical data. Providing the quantitative values underlying the phase diagrams and kinetic analyses would be essential for reproducibility and for assessing data quality.
- The level of characterisation is insufficient to support the claim that bona fide LLPS condensates are formed. Bright-field microscopy, DLS, and FRAP provide morphological information but do not establish the molecular organisation or Zn²⁺ coordination environment. Complementary spectroscopic data (e.g., circular dichroism, FTIR, or NMR) would be required to confirm peptide structure and metal binding. Without these, it remains uncertain whether the observed droplets are genuine dynamic condensates or partially aggregated assemblies.
- The kinetic studies rely on a single substrate (4-MU-acetate), and the comparison to literature values obtained with other substrates (e.g., p-nitrophenyl acetate) is not meaningful. The mechanistic claims rely entirely on computational models, without experimental validation through pH-rate profiles, isotope effects, or metal titrations. These additional experiments would be essential to establish whether the observed catalysis indeed proceeds through the proposed Zn²⁺-dependent and Zn²⁺-independent pathways.
- The manuscript would benefit from a clearer structure separating experimental results from interpretation. The text is occasionally difficult to follow, and figure descriptions are fragmented between the main text and the Supplementary Information. The inclusion of peptide sequences or schematic representations in the main figures (currently confined to Table S1) would greatly improve readability.

I also evaluated the responses to Reviewer 3's comments. I can confirm that the comments raised by Reviewer 3 have been satisfactorily addressed.

Version 2:

Reviewer comments:

Reviewer #1

(Remarks to the Author)

The authors have done a terrific job doing the revision. Congratulation for this beautiful work.

Reviewer #4

(Remarks to the Author)

The revised manuscript has been significantly improved compared to the original version. The authors have further strengthened the discussion to more clearly articulate the conceptual contribution of the work, including the intrinsic catalytic properties of the peptide condensates and the Zn(II)-dependent switching between two distinct catalytic regimes, which is now also reflected in the revised title. In addition, the revised manuscript includes substantial new experimental data, including expanded kinetic analyses and improved condensate characterization, together with additional controls that help clarify the physicochemical basis of the observed reactivity.

Overall, the authors have carefully addressed my comments I believe that the manuscript is now suitable for publication.

Point-by-point Responses to Reviewers comments:

Reviewer #1

This paper demonstrated that: condensates formed by minimalistic histidine-containing peptides can catalyze ester hydrolysis via two pathways: a Zn^{2+} -dependent mechanism activating bound water and a metal-free mechanism driven by LBHB. The concept is of highly interested and the discovery is exciting. However, the authors want to address the following technical considerations before this can be published.

1. This work suffers from a fundamental flaw, namely the overlooking of the effect of pH on hydrolysis reactions. It is well known that pH affects hydrolysis reactions. Recently, more and more studies have shown that phase separation can create a distinct pH within condensates (see, doi.org/10.1101/2024.05.23.595321). The mechanism underlying this distinct pH environment is the partitioning of ions (including protons) during LLPS. In this study, the authors tested (1) condensates made of different peptides and (2) condensates formed in the presence of Zn^{2+} or other ions, both conditions that can create distinct intra-condensate pH. Therefore, the authors should first demonstrate that

(1) condensates made of different peptides possess the same (or at least similar) pH;
(2) condensates formed at different Zn^{2+} concentrations possess the same (or at least similar) pH.

Intra-condensate pH can be measured with fluorescent probes.

(3) study how distinct pH can modulate the reactions.

These experiments will provide alternative explanations to the current observations.

Response: We thank the reviewer for the overall positive feedback and for the constructive and important comment.

We agree that pH is a critical factor for hydrolysis reactions and that measuring the pH in the dilute vs dense phase of the different condensate systems in this study can provide new insights.

To address the comment, we have followed the study mentioned by the reviewer (Knowles and coworkers bioRxiv) as well as studies by Arosio (Nat. Commun. 2025) and Chilkoti and coworkers (*Chem* 9, no. 6 (2023): 1594-1609) and performed emission spectroscopy and CLSM analysis using the pH probe SNARF-1. For this, we first analyzed how buffer pH affects the ratio between the two emission maxima of the probe, at $\lambda_{em}=580$ nm and $\lambda_{em}=640$. This results in presented in Figure S8 of the revised SI.

Next, we used CLSM to create calibration curves for each buffer by measuring the fluorescence intensity of SNARF-1 using two channels, $\lambda_{em}=580$ nm and $\lambda_{em}=640$. The various calibration curves included Tris buffer without $ZnCl_2$ and with 0.2, 0.33, and 0.67 mM of $ZnCl_2$. We next analyzed the I_{580nm}/I_{640nm} ratio of R2H/E2H condensates formed with $ZnCl_2$ at varying concentrations. We found that the dense phase of all three condensates is more basic than the dilute phase, where the dense-phase pH of condensates formed with the lower $ZnCl_2$ concentration (0.2 mM and 0.33 mM) found to be >9 and more basic compared to that with the highest $ZnCl_2$ (0.67 mM). We attribute the difference in dense-phase pH at low and high Zn^{2+} concentration to Zn^{2+} -His coordination. At lower Zn^{2+} , fewer His residues are coordinated, increasing the effective basicity of the dense phase. At higher Zn^{2+} , extensive Zn^{2+} -His coordination reduces the availability and basicity of

free His residues, resulting in a lower dense-phase pH. The elevated basicity observed at low Zn^{2+} is expected to promote the hydrolytic reaction. For the Zn^{2+} -free system, both the dilute and the dense phases of R2H/E and R/E were found to be basic, and no significant differences between the systems. To test how basicity affects the reaction, we analyzed the kinetics of hydrolysis at pH 7.5 and 9.0 (presented in Fig. S10 and S15). Indeed, the basic pH accelerated the kinetics of the reaction.

For two systems, R2H/E2H at 0.33 mM Zn^{2+} and R2H/E we were unable to determine the exact dense-phase pH value as the 580/640 nm intensity ratio of these systems lies outside the range of the fitting model equation applied for the calibration curves. Yet the results indicate that these pH values are > 9.0 .

The pH analysis was included in Fig. 2 and Fig. 5 and the values are presented in Table S6 of the SI. The following text was added to the MS (page 8 line 5): “To further elucidate how Zn^{2+} concentration influences the catalytic activity of the phase-separated systems, we measured the pH of both the dilute and dense phases using the ratiometric pH probe SNARF-1 and by confocal laser scanning microscopy (CLSM)^{47,53,54}. The emission ratio of SNARF-1 at $\lambda_{\text{em}} = 580$ and 640 nm varies as a function of pH (Fig. S8). Calibration curves were obtained by measuring the $I_{580/640}$ ratio in buffers of defined pH containing varying Zn^{2+} concentrations, enabling quantitative determination of pH in each phase (Fig. 2f–g). The pH of the dilute phase was similar across all systems, with 7.7 ± 0.11 and 7.6 ± 0.40 for 0.2 mM and 0.67 mM Zn^{2+} , respectively. In contrast, the dense-phase pH was strongly dependent on Zn^{2+} concentration, decreasing from a highly basic value of 10.8 ± 0.60 at 0.2 mM Zn^{2+} to 8.5 ± 0.10 at 0.67 mM Zn^{2+} (Fig. 2f–g, Table S6).”

Page 9 line 13: “The difference in dense-phase pH at varying Zn^{2+} concentration can be attributed to Zn^{2+} -His coordination. At lower Zn^{2+} concentrations, fewer His residues are coordinated, increasing the effective basicity of the dense phase. At higher Zn^{2+} concentrations, extensive Zn^{2+} -His coordination reduces the availability and basicity of free His residues, resulting in a lower dense-phase pH. The elevated basicity observed at low Zn^{2+} is expected to promote the hydrolytic reaction, consistent with enhanced reaction rate measured in bulk buffer under basic conditions (Fig. S10) and provides a mechanistic basis for the highest catalytic activity observed at 0.2 mM Zn^{2+} .”

Page 15 line 21: “To shed light on the catalytic capacity of the condensates, we measured the pH of the dense and dilute phases using the ratiometric probe SNARF-1. Both phases in the R2H/E and R/E systems were found to be basic, with dilute-phase pH values of 8.0 ± 0.1 and 8.4 ± 0.3 , respectively, and no significant differences between the systems (Fig. 5d–e, Table S6). The basic environment in both phases is expected to promote the hydrolytic reaction, consistent with the enhanced catalytic activity observed in bulk buffer under basic conditions (Fig. S15).”

These new insights are also reflected in the revised Discussion and Abstract.

2. If, as is very likely, condensates made of different peptides possess different pH values, how do the authors explain the mechanisms underlying the observed hydrolytic activities?

Response: The overall basic pH found in all systems is expected to increase the efficiency of the reaction as detailed in the response to the previous comment (and as

presented in Fig. S10 and Fig. S15). As mentioned above, we did not find a significant difference between the dilute and dense-pH values of different peptides (R2H/E vs. R/E).

As histidine is a titratable amino acid, its protonation behavior offers valuable insights into the unique local acid-base variations characteristic of the droplet. We modulated the protonation states of histidine residues within the R2H/E system to varying degrees in MD simulations. Considering that the pKa for the complete deprotonation of His to form its anionic form is approximately 13, we only consider its neutral and cationic forms. The detailed information can be found in Table S9.

As shown in Figure S22, we compared the collapse degree and the pairwise histidines' Ca-Ca distance distribution across neutral and partially protonated systems. The results demonstrate that histidine state did not significantly alter the LLPS properties of the peptide droplet (degree of collapse). Furthermore, the Ca distance distribution remained largely unaffected, suggesting that the structural prerequisites for LBHB formation are maintained. These results suggest that while the local acid-base environment within the droplet may differ from that of the bulk phase, our proposed molecular mechanism remains relatively robust to pH variations. This stability might be attributed to histidine partially acting as a buffer agent, effectively maintaining the internal structural integrity of the LLPS under varying solution conditions.

The results of the simulation are presented in Fig. S22 and Table S9 and the connection to the catalytic mechanism was clarified in the text as follows (page 18 line 5): “Lastly, we modulated the protonation states of His residues within the R2H/E system to varying degrees in MD simulations (Table S9). Considering that the pKa for the complete deprotonation of His to form its anionic form is approximately 13, we only consider its neutral and cationic forms. We compared the collapse degree and the pairwise His' Ca-Ca distance distribution across neutral and partially protonated systems (Fig. S22). The results demonstrate that His state does not significantly alter the LLPS properties of the peptide droplet (degree of collapse). Furthermore, the Ca distance distribution remained largely unaffected, suggesting that the structural prerequisites for LBHB formation are maintained. These results suggest that while the local acid-base environment within the droplet may differ from that of the bulk phase, our proposed molecular mechanism remains relatively robust to pH variations. This stability might be attributed to histidine partially acting as a buffer agent, effectively maintaining the internal structural integrity of the LLPS under varying solution conditions.”

3. The authors showed that condensates formed in the presence of Zn²⁺ become more compact, which in turn lowers their hydrolytic capability. However, more compact condensates may also be more hydrophobic inside, which would facilitate the partitioning of low-polarity molecules such as pNPA or 4-methylumbelliferyl acetate. Therefore, the authors should directly demonstrate the limited partitioning of 4-methylumbelliferyl acetate, for example by directly using 4-MU.

Response: We thank the reviewer for the comment. To directly address this point, we quantified the encapsulation efficiency (EE) of the 4-MU product in R2H/E2H condensates formed at varying ZnCl₂ concentrations by measuring the product concentration remaining in the dilute phase using absorbance spectroscopy with a calibrated standard. Contrary to the expectation that increased compaction would enhance partitioning of low-polarity molecules, we observe an inverse correlation

between Zn^{2+} concentration and 4-MU recruitment. Specifically, condensates formed at lower Zn^{2+} concentrations (0.2 and 0.33 mM) exhibit significantly higher 4-MU EE compared to those formed at 0.67 mM Zn^{2+} (Fig. S9).

We attribute this trend to Zn^{2+} -mediated crosslinking within the dense phase at higher Zn^{2+} concentrations, which reduces internal diffusivity and might limit accessible free volume and porosity. Thus, although higher Zn^{2+} leads to more compact condensates, this compaction does not promote product partitioning; instead, it restricts molecular uptake and contributes to the reduced catalytic performance observed under these conditions.

The results of the EE analysis are presented in Fig. S9 of the revised SI. The following text was included in the revised MS (page 9 line 10): “The partitioning of the 4-MU product was weakest in condensats formed at 0.67 mM Zn^{2+} (Fig. S9), likely due to the highly crosslinked dense-phase network at this Zn^{2+} concentration, which reduces accessibility and limits internal porosity.”

4. Using rhodamine B to probe partitioning and diffusion in this coacervate system can be problematic. Rhodamine B is a charged molecule, so its partitioning and diffusion can be affected by electrostatic interactions with the charged peptides. Because the hydrolysis substrates used in this article are non-charged, the authors should replace rhodamine B with a neutral fluorescent molecule.

Response: We agree with the reviewer that the ionizable groups of rhodamine B may promote electrostatic interactions with the charged peptides and thus influence its diffusion within the dense phase. Nevertheless, rhodamine B is widely used in the LLPS literature for FRAP measurements and is predominantly zwitterionic at neutral pH, containing both positive and negative charges that may engage in opposing (attractive and repulsive) interactions. More broadly, most commonly used FRAP probes in biomolecular condensates, including GFP and its derivatives, also carry net charge.

To directly address the reviewer’s concern, we performed additional FRAP experiments using two neutral fluorescent dyes: NBD-hexylamine and BODIPY. As shown in Fig. 1a, both dyes exhibited extremely weak fluorescence in the dense phase and pronounced inhomogeneous distribution across the sample, particularly for BODIPY. We attribute this behavior to non-specific adsorption to the Pluronic-coated wells and limited partitioning within the condensates.

Despite these limitations, FRAP recovery curves were obtained and the corresponding $t_{1/2}$ values were extracted (Fig. 1b). While the absolute $t_{1/2}$ values for both neutral dyes are significantly lower than those obtained with rhodamine B, the overall trend is preserved. Increasing the Zn^{2+} concentration from 0.33 mM to 0.67 mM results in a 2.3-fold and 3-fold increase in $t_{1/2}$ for NBD-hexylamine and BODIPY, respectively, consistent with the trend observed using rhodamine B.

We therefore conclude that the higher absolute $t_{1/2}$ values obtained with rhodamine B likely reflect stronger dye–peptide interactions in the dense phase. However, due to the extremely weak signal and non-specific distribution of the neutral dyes, rhodamine B provides more reliable FRAP data in this system. To explicitly acknowledge this limitation, we have added the following text to the revised manuscript (page 8, line 19): “We note

that the absolute diffusivity values obtained using rhodamine B may be influenced by electrostatic dye–peptide interactions.”

Fig. 1. a. Representative CLSM images of R2H/E2H condensates formed at 0.33 and 0.67 mM ZnCl₂ and loaded with rhodamine B (0.1 μM, dissolved in DDW), NBD-hexylamine (10 μM, dissolved in MeCN), and BODIPY (1.5 μM, dissolved in MeCN). Scale bars = 2 μm. **b.** t_{1/2} values obtained based on recovery plots of FRAP analysis of R2H/E2H condensates at 0.33 and 0.67 mM ZnCl₂. Data are presented as mean ± STD, number of analyzed droplets (n) is indicated.

5. The authors investigated the mechanism underlying the Zn²⁺-independent hydrolytic activity of the coacervates and attributed it to LBHB formation. However, evidence for LBHBs within the Zn²⁺-containing condensates is lacking. Since Zn²⁺ makes the condensates more compact, it could, in principle, promote LBHB formation and thus enhance the reaction — which would contradict the authors’ conclusion that Zn²⁺ inhibits the hydrolytic capability of the coacervates. The authors should clarify this point.

Response: While Zn²⁺ binding does increase the overall density of the condensate, the specific interaction between Zn²⁺ and histidine residues draws histidines away from each other, reducing the likelihood of the close proximity and optimal orientation required for efficient LBHB formation between histidines. Therefore, the mechanism promoting compactness (Zn²⁺-His coordination) and the mechanism required for LBHB formation (His-His interactions) are applicable under different environmental conditions. This competition between Zn²⁺-histidine coordination and histidine-histidine interactions clarifies the apparent contradiction raised by the reviewer. It explains why the presence of Zn²⁺, despite leading to a more compact structure, results in diminished hydrolytic activity, consistent with our conclusion that Zn²⁺ inhibits the LBHB-dependent pathway.

6. For experimental rigor, please provide mass spectrometry data verifying the occurrence of hydrolysis reactions.

Response: We thank the reviewer for the comment. To verify the occurrence of ester hydrolysis, we analyzed reaction mixtures from all catalytic systems at t=30 min using high-resolution mass spectrometry (HR-MS). In all cases, we detected a signal corresponding to the expected hydrolysis product (m/z≈177.06), confirming product formation. Notably, the same ion was also observed in the substrate-only control, which we attribute to partial hydrolysis occurring during the MS ionization process, as well as to the higher ionization efficiency of the product relative to the ester substrate.

While HR-MS therefore confirms the identity of the reaction product, it does not provide quantitative information on reaction extent. Consequently, quantitative assessment of catalytic activity was performed using time-resolved fluorescence spectroscopy, which allows direct, sensitive, and reproducible monitoring of product formation in intact two-phase systems. These analyses are presented in Fig. 2 and Fig. 5 and summarized in the corresponding Supplementary Information tables. We included the MS data is presented in the revised SI Fig. S6 and Fig. S13.

Reviewer #2

The revised manuscript by Massarano et al. has been significantly improved compared to the original version in terms of both novelty and experimental rigor. First, the authors have reinforced the discussion part to clearly address the conceptual novelty that their work has demonstrated, including intrinsic catalytic properties and Zn-dependent switching between two distinct mechanisms of catalysis. (which is now also conveyed in the revised title). Additionally, the authors have included significant amount of rigorous kinetic studies, in addition to the FRAP and DLS experiments that were originally missing, to demonstrate the sequence-based control over the condensate catalytic and material characteristics.

Therefore, the authors have answered all of my points and concerns provided in the first round of review. I believe that the manuscript is now ready for the publication, and thus recommend it to be accepted. I believe that this work will provide a novel addition to the research field of LLPS, and improve our understanding of how condensate properties could be precisely tuned in LLPS-based systems.

Response: We sincerely thank the reviewer for the positive assessment of the revised manuscript. We are grateful for the recognition of the conceptual advances and overall suitability of the work for publication.

Reviewer #4

The manuscript by Lampel and co-workers describes the formation of LLPS peptide condensates capable of catalysing ester hydrolysis through either Zn²⁺-dependent or Zn²⁺-independent mechanisms. The authors combine microscopy, kinetic measurements, and computational studies to explore how histidine residues and metal coordination contribute to catalysis.

Overall, the manuscript lacks a clear hypothesis-driven design, quantitative experimental validation of the proposed mechanisms, and the level of mechanistic generality expected for the journal.

Considering the level of novelty of the topic and the scope of the study, I do not believe that the manuscript is suitable for publication in Nature Communications, which addresses a broader multidisciplinary audience. A more specialised chemistry journal focusing on peptide-based catalysis or supramolecular systems would likely be a more appropriate venue.

Response: See response below to comment #1.

1. The manuscript focuses on peptide LLPS systems and their catalytic activity toward ester hydrolysis. Although this topic remains of chemical interest, it does not represent

a particularly timely or conceptually novel advance for the Nature portfolio. Several studies have already reported catalytic coacervates and peptide condensates employing similar model reactions. The present work refines rather than expands the existing framework and therefore does not meet the broad conceptual impact expected for this journal.

Response: We thank the reviewer for this comment and respectfully disagree with the assessment that the present work represents a purely incremental refinement of existing studies. While catalytic activity in coacervates and peptide condensates has been reported previously, the unique objective of our study is to dissect how catalytic performance in peptide condensates emerges from the interplay between peptide sequence, metal coordination, condensate microenvironment, and internal dynamics, rather than from phase separation alone.

We acknowledge that this distinction may not have been sufficiently clear in the original version of the manuscript. However, our extensive dataset, presented in both the main text and the SI, provides a comprehensive characterization of each system, including phase behavior, material properties, Zn^{2+} accumulation, internal mobility, and fully quantitative kinetic analyses with appropriate controls. Importantly, the study establishes two distinct catalytic regimes within the same condensate platform, enabling switching between a Zn^{2+} -independent, histidine-driven mechanism and a Zn^{2+} -dependent regime in which metal coordination modulates catalysis through multiple coupled effects.

To further address this conceptual point, and in response to comments from Reviewer 1, we have added substantial new analyses to the revised manuscript. Specifically, we quantified the pH of both the dilute and dense phases across peptide sequences and Zn^{2+} concentrations. We found that the dense phase of all condensates is more basic than the dilute phase, with condensates formed at lower ZnCl_2 concentrations (0.2 mM) exhibiting dense-phase pH values >9 , compared to lower pH at the highest ZnCl_2 concentration (0.67 mM). We attribute this trend to Zn^{2+} -His coordination: at low Zn^{2+} , fewer histidine residues are coordinated, increasing the effective basicity of the dense phase, whereas extensive coordination at higher Zn^{2+} reduces histidine basicity and lowers dense-phase pH. The elevated basicity observed at low Zn^{2+} is expected to promote hydrolytic reactions.

For the Zn^{2+} -free systems, both the dilute and dense phases of R2H/E and R/E were found to be basic, with no significant differences between them. To directly test the role of basicity, we performed kinetic measurements of ester hydrolysis at pH 7.5 and 9.0 (Fig. S10 and S15), which confirmed that increased pH accelerates the reaction. Together with FRAP-derived diffusivity measurements and Zn partitioning analyses, these results demonstrate that catalytic output is governed by interdependent effects of dense-phase basicity, molecular mobility, and metal coordination, rather than hydrophobicity or compaction alone. To highlight this important point, we created a schematic framework in **Fig. 1c** showing how the different properties of the condensates affect their catalytic activity (please see response to comment #6). In addition, we reflect these insights in the revised **Discussion**.

We therefore believe that the present work extends the existing framework of catalytic LLPS systems from descriptive observations to a mechanistic and quantitative

understanding, establishing sequence-encoded and metal-regulated design principles for catalytic condensates. On this basis, we therefore believe that the study provides insights of broad relevance to biomolecular condensates, supramolecular catalysis, and emergent chemical function in soft matter systems.

2. The rationale behind the experimental sequence requires clearer justification. The authors initially state that Zn^{2+} promotes LLPS and contributes to catalysis, but the data indicate that higher Zn^{2+} concentrations inhibit activity. This outcome is intriguing yet undermines the central design logic. It remains unclear why the system was optimised around Zn^{2+} if Zn^{2+} coordination ultimately suppresses reactivity. The authors should explicitly address whether the Zn^{2+} -free system was conceived as a control or as an alternative catalytic regime.

Response: We thank the reviewer for this comment and agree that the experimental rationale required clarification. Our initial hypothesis, based on prior work on histidine-containing catalytic peptides, was that Zn^{2+} coordination would enhance ester hydrolysis while promoting LLPS. While Zn^{2+} indeed promotes condensate formation, we unexpectedly found that higher Zn^{2+} concentrations inhibit catalysis by inducing His- Zn^{2+} -mediated crosslinking, reducing internal diffusivity and limiting product (and likely also substrate) partitioning.

This unexpected behavior motivated a focused mechanistic investigation into how metal coordination can simultaneously promote phase separation while suppressing catalysis. Importantly, the Zn^{2+} -free system was conceived not merely as a control, but as an alternative catalytic regime that isolates histidine-driven chemistry from metal-mediated effects. To clarify this point, we have revised the manuscript to explicitly frame the Zn^{2+} -free system as a complementary mode of catalysis.

To clarify this point we revised as follows:

Page 6 line 5: “Building on previous work on esterase-mimicking catalytic peptides¹⁶, we used Zn^{2+} as a cofactor for ester hydrolysis and hypothesized that Zn^{2+} could function both as a cofactor and an initiator of LLPS (Fig. 1).”

Page 12 line 10: “In light of the inhibitory effect of Zn^{2+} on condensate catalytic capacity, we next examined catalysis in the absence of ZnCl_2 as an alternative mechanistic regime. We first analyzed the LLPS propensity of oppositely charged peptides (at 1:1 ratio) in the absence of ZnCl_2 ...”

3. Some figure references are inconsistent or incomplete (e.g., “Fig. 1b–g” does not correspond to any figure panels). Furthermore, key datasets are represented only as heat maps or schematic diagrams, with no access to the raw numerical data. Providing the quantitative values underlying the phase diagrams and kinetic analyses would be essential for reproducibility and for assessing data quality.

Response: As indicated in the original manuscript, all raw numerical data underlying the kinetic analyses were deposited in Zenodo and are accessible through the link provided at the end of the manuscript. To further improve transparency and reproducibility, we have now included the quantitative turbidity values underlying the phase-diagram heat maps in Tables S2–S4 of the revised SI. In addition, we have carefully checked and corrected all figure references throughout the manuscript to ensure consistency and accuracy.

4. The level of characterisation is insufficient to support the claim that bona fide LLPS condensates are formed. Bright-field microscopy, DLS, and FRAP provide morphological information but do not establish the molecular organisation or Zn^{2+} coordination environment. Complementary spectroscopic data (e.g., circular dichroism, FTIR, or NMR) would be required to confirm peptide structure and metal binding. Without these, it remains uncertain whether the observed droplets are genuine dynamic condensates or partially aggregated assemblies.

Response: We thank the reviewer for this constructive comment. In the LLPS field, bona fide condensates are defined primarily by their dynamic behavior, and the most widely accepted criteria are the observation of droplet coalescence and the presence of internal molecular mobility measured by FRAP. These functional characteristics, rather than a specific spectroscopic signature, define condensates, as established in foundational studies from the laboratories of Pappu, Rosen, and Brangwynne [For reference see: Dar, F., et al, 2024, *Nature communications*, 15(1), 3413; Gibson, B. A. et al, 2019, *Cell*, 179, 470-484; Gibson, B.A. and Rosen, M.K., 2023. *PNAS*, 120, p.e2218085120; Bracha, D. et al, *Nature biotechnology*, 37(12), 1435-1445].

To address the reviewer's concern directly, we performed additional CLSM experiments and now include time-lapse videos (Supplemental Videos 1-5) that clearly show fusion and coalescence of droplets across all peptide systems. These movies visibly demonstrate the fluid nature of the assemblies and confirm that they are not partially aggregated particles but dynamic liquid condensates. The FRAP measurements included in the manuscript further support this conclusion by revealing substantial recovery of fluorescence after photobleaching for both Zn^{2+} -dependent and Zn^{2+} -free condensates, consistent with a mobile internal environment.

Beyond establishing liquid behavior, we also expanded the molecular characterization of peptide structure and Zn^{2+} coordination by performing CD analysis. The CD analysis shows that the His-containing peptides retain a random-coil signature, yet exhibit distinct spectral shifts upon addition of $ZnCl_2$. For E2H, a new maximum near 203 nm appears and intensifies with increasing Zn^{2+} concentration, while R2H displays a progressive decrease in its random-coil minimum under the same conditions. In contrast, peptides lacking His show no detectable spectral changes in the presence of Zn^{2+} , indicating that the metal interacts specifically with His-containing peptides. These observations are fully consistent with Zn^{2+} -His coordination contributing to LLPS and to the altered material properties of the condensates.

We also expanded our Trp emission analysis, which reinforces these findings. His-containing peptides show strong Zn^{2+} -induced increase in Trp fluorescence intensity, whereas peptides without His do not. The agreement between CD and Trp emission results provides an independent line of evidence that Zn^{2+} binds selectively to His-bearing peptides and supports the mechanistic interpretation that Zn^{2+} modulates both condensate formation and the internal dynamics captured by FRAP.

Taken together, the coalescence videos, FRAP data, CD spectra, and Trp emission analyses demonstrate that the assemblies are phase separated condensates. These additional datasets directly address the reviewer's concern regarding the level of characterization and establish both the liquid nature of the droplets and the molecular basis of Zn^{2+} interaction within the system.

The results of the CD and Trp emission analyses are presented in Fig. S12 of the revised SI. The following text was added to the revised MS (page 11 line 14): “To confirm that the peptide directly interacts with Zn^{2+} ions, we performed CD analysis in the absence or presence of $ZnCl_2$. The CD analysis (Fig. S12) shows that the His-containing peptides R2H and E2H retain a random-coil signature with a ~ 196 nm minimum and a ~ 225 nm maximum, yet show distinct spectral shifts upon addition of $ZnCl_2$. For E2H, a new maximum near 203 nm appears and intensifies with increasing Zn^{2+} concentration, while R2H displays a progressive decrease in its random-coil minimum under the same conditions. In contrast, peptides lacking His show no detectable spectral changes in the presence of Zn^{2+} , indicating that the metal interacts specifically with His-containing peptides. In addition, we leveraged the intrinsic fluorescence of Trp within the peptides to perform Trp emission analysis in which environmental changes to the indole side chain results in changes in the emission intensity ($\lambda_{ex}=280$ nm). The results show a significant increase in the Trp emission intensity for the His-containing peptides, R2H and E2H, in the presence of 0.67 mM $ZnCl_2$ while no change is observed for the peptides lacking His, R and E (Fig. S12).

Together, the CD and Trp emission analysis show that Zn^{2+} ions interact with the His-containing peptides. These observations strengthen the phase diagrams and FRAP results which show that Zn^{2+} promotes phase separation and slows the dense phase internal mobility, likely by strong interactions with R2H and E2H.”

5. The kinetic studies rely on a single substrate (4-MU-acetate), and the comparison to literature values obtained with other substrates (e.g., p-nitrophenyl acetate) is not meaningful. The mechanistic claims rely entirely on computational models, without experimental validation through pH-rate profiles, isotope effects, or metal titrations. These additional experiments would be essential to establish whether the observed catalysis indeed proceeds through the proposed Zn^{2+} -dependent and Zn^{2+} -independent pathways.

Response: Please see response to comment #1 and responses to Reviewer 1 comments #1 and #2.

6. The manuscript would benefit from a clearer structure separating experimental results from interpretation. The text is occasionally difficult to follow, and figure descriptions are fragmented between the main text and the Supplementary Information. The inclusion of peptide sequences or schematic representations in the main figures (currently confined to Table S1) would greatly improve readability.

I also evaluated the responses to Reviewer 3's comments. I can confirm that the comments raised by Reviewer 3 have been satisfactorily addressed.

Response: We thank the reviewer for the comment. To address the point, we have revised the text throughout the manuscript to separate the results from their interpretation (changes highlighted in red in the MS). Wherever suitable, we dedicated the last paragraph of every section in the manuscript for interpretation of the results and a short discussion of the findings.

In addition, following the reviewer's comment, we have revised Figure 1 to show the peptide sequences. Moreover, following the other comments raised by the reviewer, we added a summarizing scheme showing the relations between the different properties of

condensates (i.e. phase separation propensity, Zn concentration, basicity, diffusivity, and partitioning) and their catalytic activity. We also reflected these insights in the abstract.